# Bridging the 3D geometrical organisation of white matter pathways across anatomical length scales and species

Hans Martin Kjer[1,2]*, Mariam Andersson[1,2], Yi He[1,3], Alexandra Pacureanu[4], Alessandro Daducci[5], Marco Pizzolato[2], Tim Salditt[6], Anna-Lena Robisch[6], Marina Eckermann[4,6], Mareike Töpperwien[6], Anders Bjorholm Dahl[2], Maria Louise Elkjær[7,8], Zsolt Illes[7,8,9,10], Maurice Ptito[2,11], Vedrana Andersen Dahl[2], Tim B Dyrby[1,2]*

[1]Danish Research Centre for Magnetic Resonance, Center for Functional and Diagnostic Imaging and Research, Copenhagen University Hospital Amager and Hvidovre, Hvidovre, Denmark; [2]Department of Applied Mathematics and Computer Science, Technical University of Denmark, Kongens Lyngby, Denmark; [3]Guangdong Provincial Engineering Research Center of Molecular Imaging, The Fifth Affiliated Hospital, Sun Yat-sen University, Zhuhai, China; [4]ESRF - The European Synchrotron, Grenoble, France; [5]Department of Computer Science, University of Verona, Verona, Italy; [6]Institut für Röntgenphysik, Universität Göttingen, Friedrich-Hund-Platz, Göttingen, Germany; [7]Department of Neurology, Odense University Hospital, Odense, Denmark; [8]Institute of Molecular Medicine, University of Southern Denmark, Odense, Denmark; [9]BRIDGE—Brain Research—Inter-Disciplinary Guided Excellence, Department of Clinical Research, University of Southern Denmark, Odense, Denmark; [10]Rheumatology Research Unit, Odense University Hospital, Odense, Denmark; [11]School of Optometry, University of Montreal, Montreal, Canada

*For correspondence:
hmkj@dtu.dk (HMK);
timd@drcmr.dk (TBD)

**Competing interest:** The authors declare that no competing interests exist.

## eLife Assessment

This **valuable** study presents new observations on white matter organisation at the micron scale, using a combination of synchrotron imaging and diffusion MRI across two species. Notably, the authors provide **solid** evidence for the fasciculation of axons within major fibre bundles into laminar structures, though these structures are not consistently observed across modalities or species. The study will be of general interest to neuroanatomists and those interested in white matter imaging.

**Abstract** We used diffusion MRI and x-ray synchrotron imaging on monkey and mice brains to examine the organisation of fibre pathways in white matter across anatomical scales. We compared the structure in the corpus callosum and crossing fibre regions and investigated the differences in cuprizone-induced demyelination in mouse brains versus healthy controls. Our findings revealed common principles of fibre organisation that apply despite the varying patterns observed across species; small axonal fasciculi and major bundles formed laminar structures with varying angles, according to the characteristics of major pathways. Fasciculi exhibited non-straight paths around obstacles like blood vessels, comparable across the samples of varying fibre complexity and demyelination. Quantifications of fibre orientation distributions were consistent across anatomical length scales and modalities, whereas tissue anisotropy had a more complex relationship, both dependent on the field-of-view. Our study emphasises the need to

balance field-of-view and voxel size when characterising white matter features across length scales.

## Introduction

The connectome is the map of axonal connections between different brain regions. Mapping of the connectome across several anatomical length scales can unravel its hierarchical organisation, which bears a close association with normal brain function and with disruptions in disease states. The major white matter pathways can be outlined at millimetre image resolution, e.g., through photographic microdissection or tractography on diffusion magnetic resonance imaging (dMRI) (*Schilling et al., 2021*). They include the association, projection, and interhemispheric pathways, which are further categorised into functionally related tracts such as the pyramidal tract and the inferior fronto occipital fasciculus. At millimetre resolution, the white matter tracts appear to form a structural backbone, but examination at higher resolution shows a particular topological and spatial organisation of the axonal fasciculi within (*Sarubbo et al., 2019*). A fasciculus is a bundle of axons that travel together over short or long distances. Its size and shape can vary depending on its internal organisation and its relationship to neighbouring fasciculi. The pyramidal tract, for example, has a topological organisation into parallel fasciculi determined by the homunculus organisation of the primary motor cortex. Hence, diseases such as amyotrophic lateral sclerosis that over time result in the loss of different types of motor control, correlate in the pyramidal tract with a topological dependent axonal degeneration (*Sach et al., 2004*).

The conventional model of white matter (WM) architecture has assumed a parallel within-tract topological organisation of fibres. Based on MRI tractography, Van Weeden proposed a sheet-like axonal organisation between tracts, with frequent right angle crossings (*Wedeen et al., 2012*), an interpretation that has been debated (*Catani et al., 2012*). Indeed, the limited spatial resolution afforded by diffusion MRI calls into question the modelling accuracy of sub-voxel fibre crossings and parallel fibres (*Maier-Hein et al., 2017*). Subsequently, *Schilling et al., 2017* explored a potential resolution limit of crossing fibres by mapping the distribution of fibre orientations from histological images using the structure tensor model. They found that even at the scale of isotropic 32 µm, voxels still contain many crossing fibres. Despite the limited resolution of dMRI, the water diffusion process can reveal microstructural geometrical features, such as axons and cell bodies, though these features are compounded at the voxel level. Consequently, estimating microstructural characteristics depends on biophysical modelling assumptions, which can often be simplistic due to limited knowledge of the 3D morphology of cells and axons and their intermediate-level topological organisation within a voxel. Thus, complementary high-resolution imaging techniques that directly capture axon morphology and fasciculi organisation in 3D across different length scales within an MRI voxel are essential for understanding anatomy and improving the accuracy of dMRI-based models (*Alexander et al., 2019*).

MicroCT from lab sources or large-scale synchrotron facilities has been demonstrated as useful for the validation of low-resolution MRI in small samples. Combining structure tensor analysis and streamlined tractography applied to the micro CT, Trinkel et al. demonstrated virtual mapping of the murine brain connectome of major pathways in comparison to tractography from diffusion MRI (*Trinkle et al., 2021*). In their work, they did not provide insights into the organisation of interfacing fasciculus bundles and were not able to resolve finer structures such as axons when using the resolution of MRI due to limited resolution.

X-ray holographic nanotomography (XNH) is another complementary imaging technique that enables nanoscopic image resolutions of intact axonal white matter in 3D using phase-contrast synchrotron imaging. By stacking image XNH volumes, *Andersson et al., 2020* obtained an anisotropic FOV of 600 micrometres, which is of comparable scale to a typical MRI voxel. Through segmentation of the larger diameter axons, they characterised the microdispersion of their trajectories using a measure of tortuosity, thereby revealing that the cross-sectional morphological changes along individual axons are imposed by structures in the local environment, i.e., cells, vacuoles, and blood vessels (*Andersson et al., 2020*). Such 3D characteristics have provided new insights that improved microstructure MRI diffusion models (*Lee et al., 2020*; *Andersson et al., 2022*). However, the Andersson study provided no analysis of the organisational features at the scale of axon fasciculus level due to the technical difficulty of segmenting full volumes. We thus see a need for a sufficiently high-resolution imaging

modality with a larger FOV for exploring and quantifying axon fasciculus organisation (especially their interfaces) in simple and complex white matter regions. In this regard, Mortazavi et al., combined a neuronal tracer with structure tensor analysis applied to histological images to show that labelled axons of the major white matter pathways in the *centrum semiovale* cross each other in a grid-pattern (*Mortazavi et al., 2018*). Moreover, *Zhou et al., 2013* combined two different fluorescence neuronal tracers to show vertical stacking of parallel interhemispheric projections in the corpus callosum of mice. However, such approaches capture only a small number of labelled axons.

Recent advances in synchrotron imaging introduced hierarchical phase-contrast CT of the intact human brain, with a selection of different FOVs and image resolution (down to 2.5 μm) within a single imaging session (*Walsh et al., 2021*). Ideally, sub-micrometre imaging techniques are needed to be comparable to the microstructural features that diffusion MRI is sensitive to.

In this work, we applied a multimodal imaging approach, aiming to bridge the gap between diffusion MRI and the scale of the axonal fasciculi, aiming to reveal the 3D white matter organisation across different anatomical length scales throughout intact monkey and mouse brains, and in a murine demyelination model. We study two white matter regions of differing organisational complexity: The relatively simple and homogenous *corpus callosum* (CC), and second a complex crossing fibre region in the *centrum semiovale* (CS). We use the term 'anatomical length scales' throughout to designate anatomical features of different magnitudes – axons, bundles of axons (fasciculi), and tracts – that may have differing scales between mouse and monkey brains. To perform this multi-scale investigation, we apply a multi-modal imaging approach combining conventional diffusion MRI with sub-micron x-ray synchrotron phase-contrast tomography at two different synchrotron facilities: the Deutsches Elektronen-Synchrotron (DESY), beamline P10; and the European Synchrotron (ESRF), beamline ID16A. After diffusion MRI, we imaged volumes of interest at DESY (550 nm voxel size) and then at ESRF (75–100 nm voxel sizes). Each step produced images with increasing resolution, but in incrementally smaller FOVs. From the diffusion MRI data, we estimated tissue anisotropy using the diffusion tensor model (*Basser et al., 1994*) and micro-tensor model (*Kaden et al., 2016*). Additionally, we modelled the fibre orientation distribution (FOD) with the multi-fibre constrained spherical deconvolution (CSD) model (*Tournier et al., 2007*). In the x-ray synchrotron data, we applied a scale-space structure tensor analysis, which allowed for the quantification of structure tensor-derived tissue anisotropy and FOD in the same anatomical regime indirectly detected by dMRI. Additionally, we performed tractography on the main (or dominant) direction obtained from the structure tensor, and modelled axonal fasciculi based on the clustering of the tractography streamlines, with quantification of their deviations from linearity, i.e., tortuosity. Through these methods, we demonstrated organisational principles of white matter that persist across anatomical length scales and species. These principles govern the organisation of axonal fasciculi into sheet-like laminar shapes (structures with a predominant planar arrangement). Interestingly, while these principles remain consistent, they result in varied structural organisations in different species. We found that these laminae have differing inclination angles with respect to each other, depending on their presence within parallel or crossing major pathways. Furthermore, using a mouse model of focal demyelination induced by cuprizone (CPZ) treatment, we investigate the inflammation-related influence on axonal organisation. This is achieved through the same structure tensor-derived micro-anisotropy and tractography streamline metrics.

## Results
### Multi-scale, multi-modal imaging data in the monkey CC

We first acquired an ex vivo diffusion MRI image of the whole monkey brain, which was reconstructed with isotropic 0.5×0.5×0.5 mm³ voxel sizes (*Figure 1A*) (see Methods section for details). We then obtained a thin needle biopsy sample of 1 mm diameter from the mid-body of the corpus callosum (CC) (*Figure 1B*), which we imaged twice with phase contrast tomography; at DESY with 550 nm voxel size, providing fascicular resolution, and at ESRF with 75 nm voxel size, providing axonal resolution. At ESRF, we stacked four consecutive image volumes to create a large effective FOV (*Figure 1C*). The relative sizes of an MRI voxel and the various synchrotron volumes are shown in *Figure 1D*. The dark image intensities in the synchrotron x-ray images (*Figure 1B and C*) are due to osmium staining of the myelin.

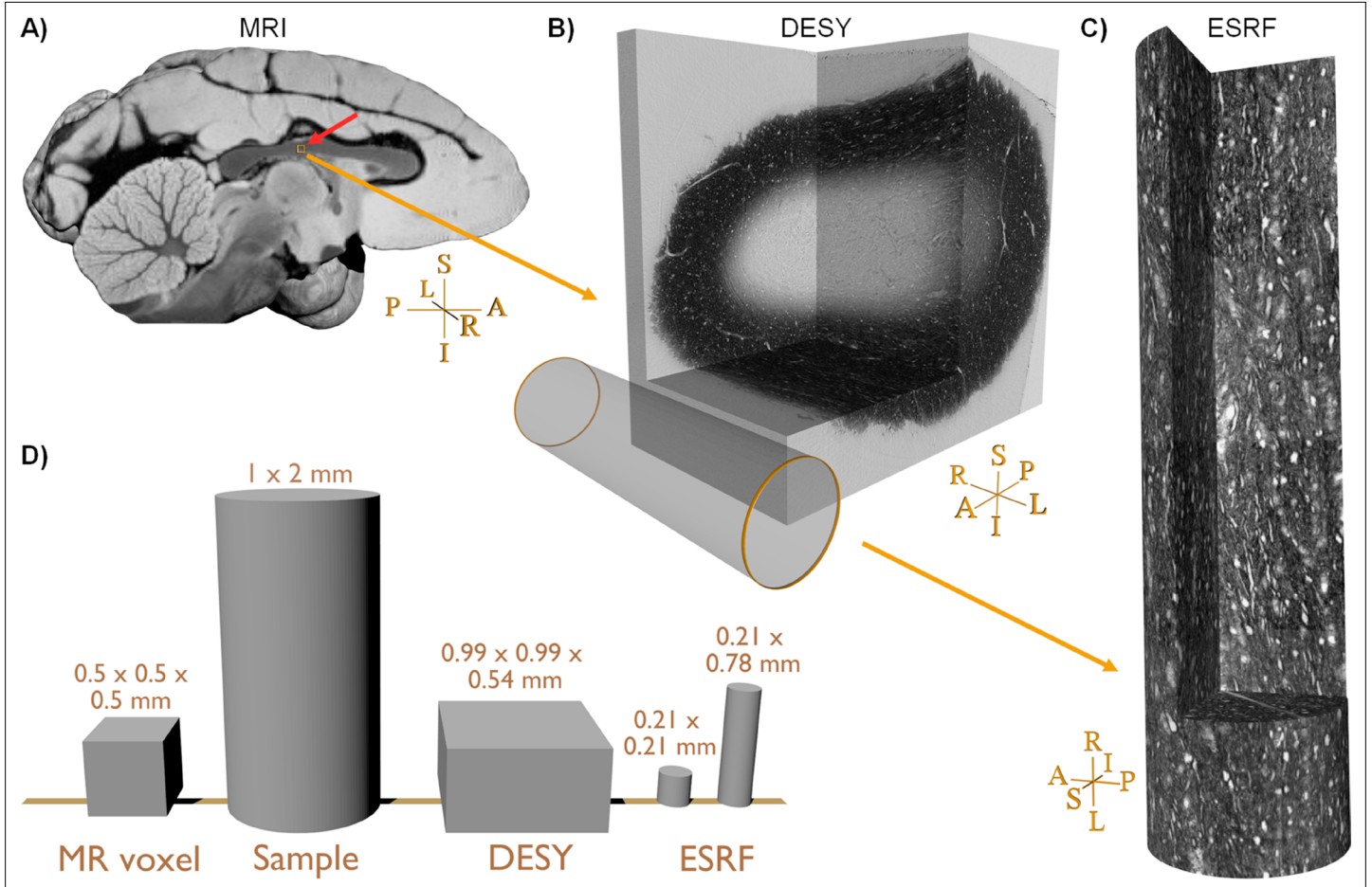

**Figure 1.** Sample comparison. (**A**) Rendering of the vervet monkey brain structural magnetic resonance imaging (MRI) close to the mid-sagittal plane. (**B**) Large field-of-view (FOV) scanning of the biopsy, Deutsches Elektronen-Synchrotron (DESY). The small orange box near the red arrow in (**A**) indicates the location and size of the FOV within the MRI data. (**C**) The stack of four small FOVs scans obtained at European Synchrotron (ESRF). The cylinder in (**B**) indicates the relative size of the FOV within the DESY scan. (**D**) Relative size comparison of one MRI voxel, the biopsy sample prior to staining and fixation, and various synchrotron FOVs. For cylindrical FOVs, the first number indicates the diameter, and the second number is the height. Sample orientations are related to the whole brain in (**A**): R: Right, L: Left, I: Inferior, S: superior, A: Anterior, P: posterior.

As seen in the image acquired at DESY in *Figure 1B*, the staining only penetrated the outer rim of the biopsy sample. At this image resolution, resolvable structures include the largest axons (with diameters roughly >2.5 µm), blood vessels, clusters of cell bodies, and vacuoles. Cell bodies and vacuoles appear bright in contrast to their outlining myelinated axons, which appeared to be oriented mainly in the left-right (L-R) axis, as expected for a CC sample.

The high-resolution ESRF data has a field-of-view (FOV) of 0.21×0.21×0.21 mm. *Figure 1B* depicts the placement of the FOV in the osmium-stained rim of the CC sample. The four stacked scans provide an extended FOV of 0.21×0.21×0.78 mm to match better that of the DESY volume and an MRI voxel (*Figure 1C*). In this data, it is possible to observe and quantify fine microstructural details, i.e., axon morphology, myelin thickness, nodes of Ranvier, clusters of cell bodies, vacuoles, and blood vessels (for a detailed description see *Andersson et al., 2020*).

## Corpus callosum: A "straight fibre" region in the monkey brain

The CC, the largest white matter tract in the brain, has high tissue anisotropy, being dominated by densely packed axon fasciculi that all run in the right-left direction between the two hemispheres (*Andersson et al., 2020*). The directional colour-coding of the three eigenvectors from the diffusion MRI tensor model suggests that the axonal directions are locally consistent and symmetric around the midsagittal plane (*Figure 2A*). The MRI voxel corresponding to the location of the synchrotron

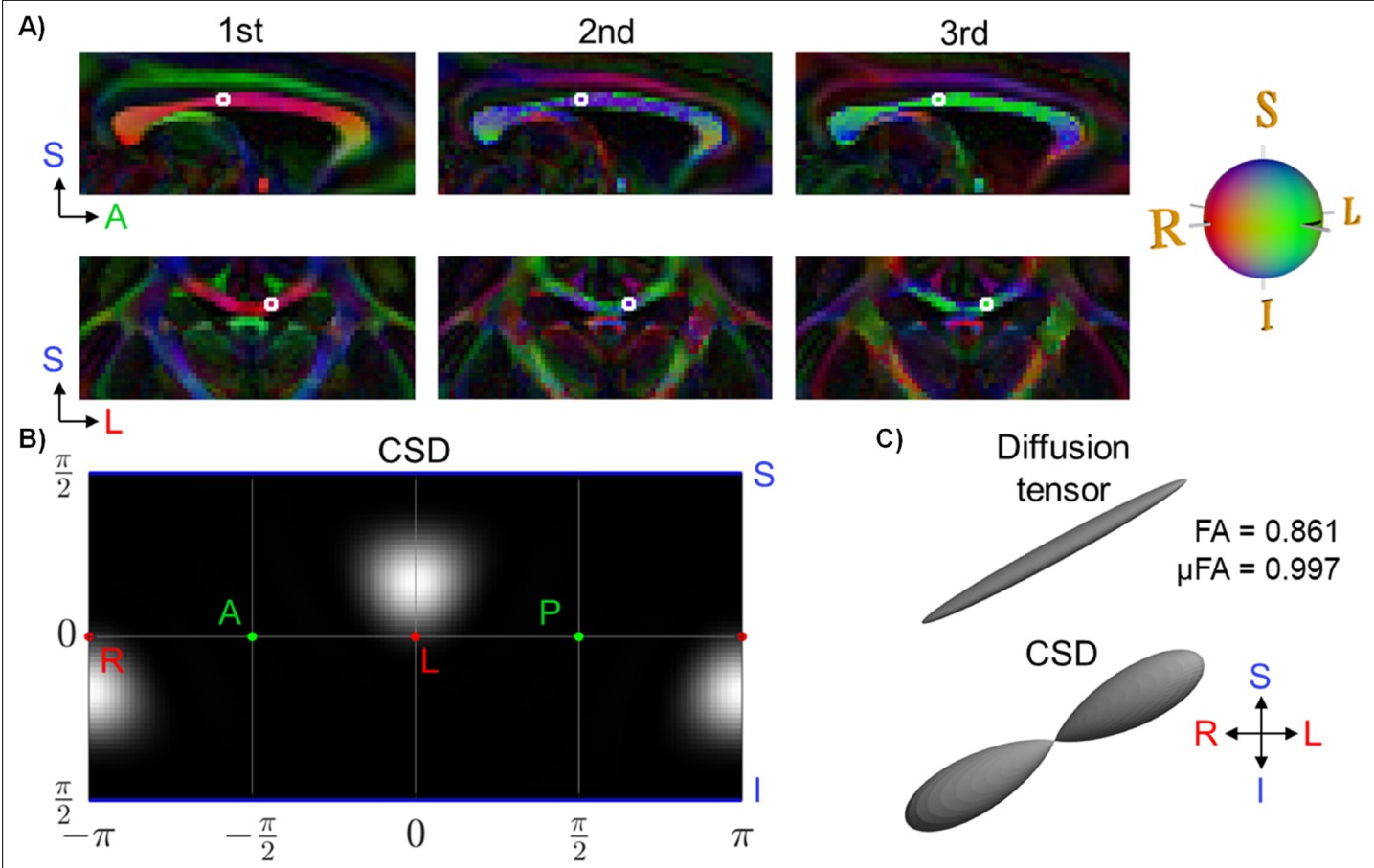

**Figure 2.** Diffusion magnetic resonance imaging (dMRI)-based orientations and tensor shapes. (**A**) The diffusion tensor model showing principal directions in a sagittal slice of the monkey corpus callosum (CC). The voxel corresponding to the biopsy sampling location (the synchrotron field-of-views, FOVs) is outlined in a white box. (**B**) The fibre orientation distribution (FOD) of the constrained spherical deconvolution (CSD) model in the selected diffusion MRI voxel is represented as a spherical polar histogram. (**C**) The corresponding glyph representations of the diffusion tensor and CSD.

samples is located in the mid-body, approximately 2 mm lateral to the midsagittal plane (the white box in *Figure 2A*). In this voxel, the estimated orientations of the diffusion tensor and CSD models (*Figure 2B and C*) both have a main R-L component (red) with a strong I-S inclination (blue), and very little response in the A-P direction (green). The multi-fibre (CSD) model shows a single fibre peak, with a relatively narrow and isotropic spread of the fibre orientation distribution (FOD) when mapped onto a spherical polar histogram (*Figure 2B*). This is consistent with a homogeneous microstructural environment dominated by densely packed parallel axons. Indeed, the diffusion tensor model predicted a high fractional anisotropy (FA) value of 0.861 (*Basser and Pierpaoli, 1996*) and even higher FA anisotropy in the micro-tensor domain of 0.997, i.e., the micro (μ)FA calculated as (*Kaden et al., 2016*).

Our analysis of the high-resolution DESY data of the CC sample is shown in *Figure 3A, C and E*. The structure tensor was estimated based on a single set of scale-space parameters, corresponding to an integration patch size of 12 microns (see methods for further details), which is sufficient to detect all tissue-relevant orientation features. Interestingly, the structure tensor directional colour map in *Figure 3A* (yellow-green bands) indicates axonal fasciculi disposed with a mainly planar organisation, thus forming laminae with a thickness of up to 40–45 μm projecting throughout the whole cross-section of the CC sample. The laminar thickness was determined by manual measurements of laminae visually identified in the 3D volume. These laminae have inclination angles up to ~35 degrees from the right-left (R-L) axis. *Figure 3C* shows a subset of the structure tensors visualised as 3D glyphs overlaid onto an image from the synchrotron volume. At this image resolution, the structure tensor directions follow either blood vessels or axonal structures.

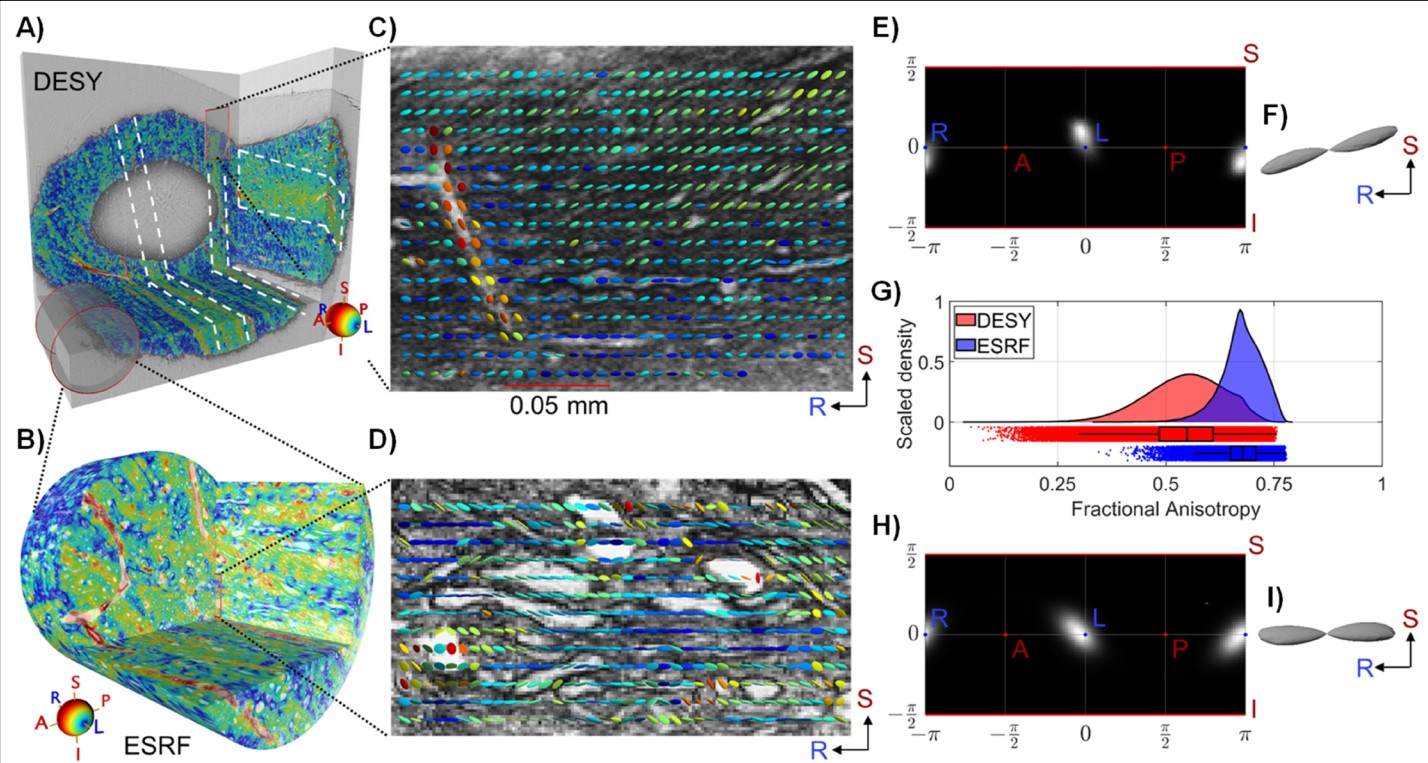

**Figure 3.** Structure tensor shape for corpus callosum (CC) sample. (**A–B**) 3D renderings from respectively the Deutsches Elektronen-Synchrotron (DESY) and European Synchrotron (ESRF) CC biopsy samples with structure tensor main direction colour coding (in accordance with the colour sphere). The white markings illustrate bands where axons are oriented at an angle compared to their surroundings, indicating a laminar organisation. The red cylinder on (**A**) shows the scale difference between the DESY and ESRF data. The position of the sample within the CC is marked on the MR image in *Figures 1A and 2A*. (**C–D**) Selected coronal slice overlaid with a regularly spaced subset of structure tensor glyphs, coloured according to their predominant direction. (**E–F**) Spherical polar histograms of the DESY structure tensor main directions (FOD) and the corresponding glyph. (**G**) Kernel density estimates of structure tensor fractional anisotropy (FA) values. (**H–I**) Spherical histogram of the ESRF structure tensor main directions (FOD) and the corresponding glyph. NB: We exclude contributions from the voxel of blood vessels in FA histograms and FODs.

Upon mapping the structure tensor principal directions within the image volume onto a spherical polar histogram (*Figure 3E*), we observe a FOD having a single fibre peak. The I-S inclination angle is smaller than that of CSD from diffusion MRI (*Figure 2B and C*). *Figure 3F* illustrates the elongated shape of the structure tensor FOD towards the I-S axis, reflecting a degree of fibre dispersion originating mainly from the axonal laminar organisation visible in *Figure 3A* (regions marked in white square).

From the ultra-high image resolution ESRF data shown in *Figure 3(B and D)*, we estimated the structure tensor at different scales, corresponding to patch sizes ranging from 9.2 to 2 µm. This allowed for the detection of tissue anisotropy on different anatomical length scales, i.e., corresponding to blood vessels, large axons, and small axons. The structure tensor directional colouring (*Figure 3B*) does not clearly capture the layered laminar organisation seen in the DESY data. This may be explained by the smaller FOV (210 µm) not covering a volume in which the laminar organisation appears.

The direction vectors of the structure tensor analysis from all four stacked ESRF volumes are converted to spherical polar coordinates in *Figure 3H* (see *Appendix 1—figure 3* for each volume individually). The histogram resembles that representing the DESY data (*Figure 3E*), except that the main direction of the ESRF data has a lower inclination angle in the I-S axis. An exact co-registration of the two synchrotron volumes was not feasible, and some discrepancies between the FODs are to be expected. The spread and shape of the FODs are quantified using the Orientation Dispersion Index (ODI) and Dispersion Anisotropy (DA) (see Methods section). For the DESY and ESRF samples, the ODI values are 0.038 and 0.069, respectively, while the DA values are 0.55 and 0.59. Despite the different FOVs, these values are similar. The low ODI indicates a high degree of axon alignment, as expected in the CC. The mid-level DA values suggest some anisotropic spread of the

directions, reflecting the angled laminar organisation observed in the DESY sample. Interestingly, the DA value for the ESRF sample is almost identical, despite the laminar bands being less visually apparent.

In *Figure 3G*, we present the calculated distribution of FA values of the structure tensor for the four stacked ESRF volumes, intended to match approximately a single MRI voxel. The mean FA value of the ESRF data was higher than in the DESY data (0.67±0.05 vs 0.54±0.09, respectively) and the ESRF data distribution was narrower than in the DESY data, i.e., (the ESRF median/Inter quartile range (IQR): 0.68/0.06, DESY median/IQR: 0.55/0.12). The high anisotropy at the micrometre-scale image resolutions is in accord with the FA value of 0.86 and µFA of 0.99 in the dMRI voxel, as shown in *Figure 2C*. Although the two synchrotron volumes originated from the same biopsy, the FA distributions are not identical. Visual inspection reveals that smaller axons and cell bodies are hardly distinguishable in the DESY data (*Figure 3A*) but are clearly evident in the higher-resolution ESRF data (*Figure 3B*).

## Deep white matter: A "complex tissue" region in the monkey brain

We next investigated the axonal organisation in the *centrum semiovale* (CS) in deep white matter. The region contains up to three major pathways that are potentially crossing, i.e., the corticospinal tract, the CC, and the superior-longitudinal fasciculus (*Schmahmann and Pandya, 2009*). We imaged this complex tissue sample with dMRI and ultra-high resolution synchrotron imaging at the ESRF, and then applied a structure tensor analysis using the same range of scale-space parameters as for the CC sample, i.e., focusing on contrasted tissue features of size in the range of 9.2–2 µm.

*Figure 4A* shows the CSD FOD multi-fibre reconstructions from dMRI of a region encompassing the sample puncture. Due to the image resolution of dMRI and the known regional complexity of fibre organisation, most voxels contained more than one fibre peak, which indicates the expected presence of crossing fibres. (*Figure 4A*) (white circle) highlights the dMRI voxel best matching the FOD in the synchrotron imaging volume; it contains three fibre components, as shown by the spherical polar histogram (*Figure 4B*) and the corresponding glyph (*Figure 4C*). The two largest peaks ('parallel fibres') are S-I directed (blue) and the smaller peaks point mostly in the R-L direction (red).

The synchrotron data from *centrum semiovale* shows axons that visually tend to organise as laminae similarly to the CC. The laminar thickness ranges from as little as 5–10 µm to as much as 35–40 µm. Some laminae are predominantly populated by axons of small diameter, too small to segment individually at this image resolution. Other laminae include a mixture of populations of different diameters. Some axons intermingle with a neighbouring lamina as shown in *Figure 4E*. The laminar findings are consistent across the four stacked sub-volumes (see the supplementary animation, *Figure 4—video 1*).

The structure tensor FOD within the four stacked ESRF volumes (*Figure 4F and G*) (see *Appendix 1—figure 3* for each volum for each volume individually) has one strong S-I and one weak R-L directed peak, much as seen in the diffusion MRI (*Figure 4B and C*). The largest discrepancy between the FODs of the two modalities lies in the presence of a secondary S-I-directed peak in the dMRI data (*Figure 4B and C*). Interestingly, visual inspection of the colour-coded structure tensor directions in *Figure 4E* shows the existence of voxels whose primary direction is along the A-P axis. However, this represents a small enough portion of the volume that it does not appear as a distinct peak on the FOD.

Estimation of ODI and DA was not performed in this case, as these indices are uninterpretable when the FOD has contributions from multiple WM pathways. It was not possible to robustly isolate the different pathways at the voxel level in this sample.

Finally, the estimated FA of the diffusion tensor within the single dMRI voxel (0.35) was much lower than that in the micro-tensor (0.99). The low FA value of the diffusion tensor model is expected as it cannot handle the complex fibre crossings (*Kaden et al., 2016*; *Jespersen et al., 2013*; *Lasič et al., 2014*), whereas the micro-tensor is modelled to be independent of the axonal organisation.

The structure tensor FA histogram from the ESRF data (*Figure 4H*) had a median value of 0.66 and of IQR 0.07. Interestingly, the structure tensor FA histograms of the 'complex region' and the 'straight fibre' regions (*Figure 3G* vs. *Figure 4H*, respectively) are almost identical, suggesting that the given scale-space structure tensor is independent of fibre organisation, similar to the diffusion micro-tensor model (µFA).

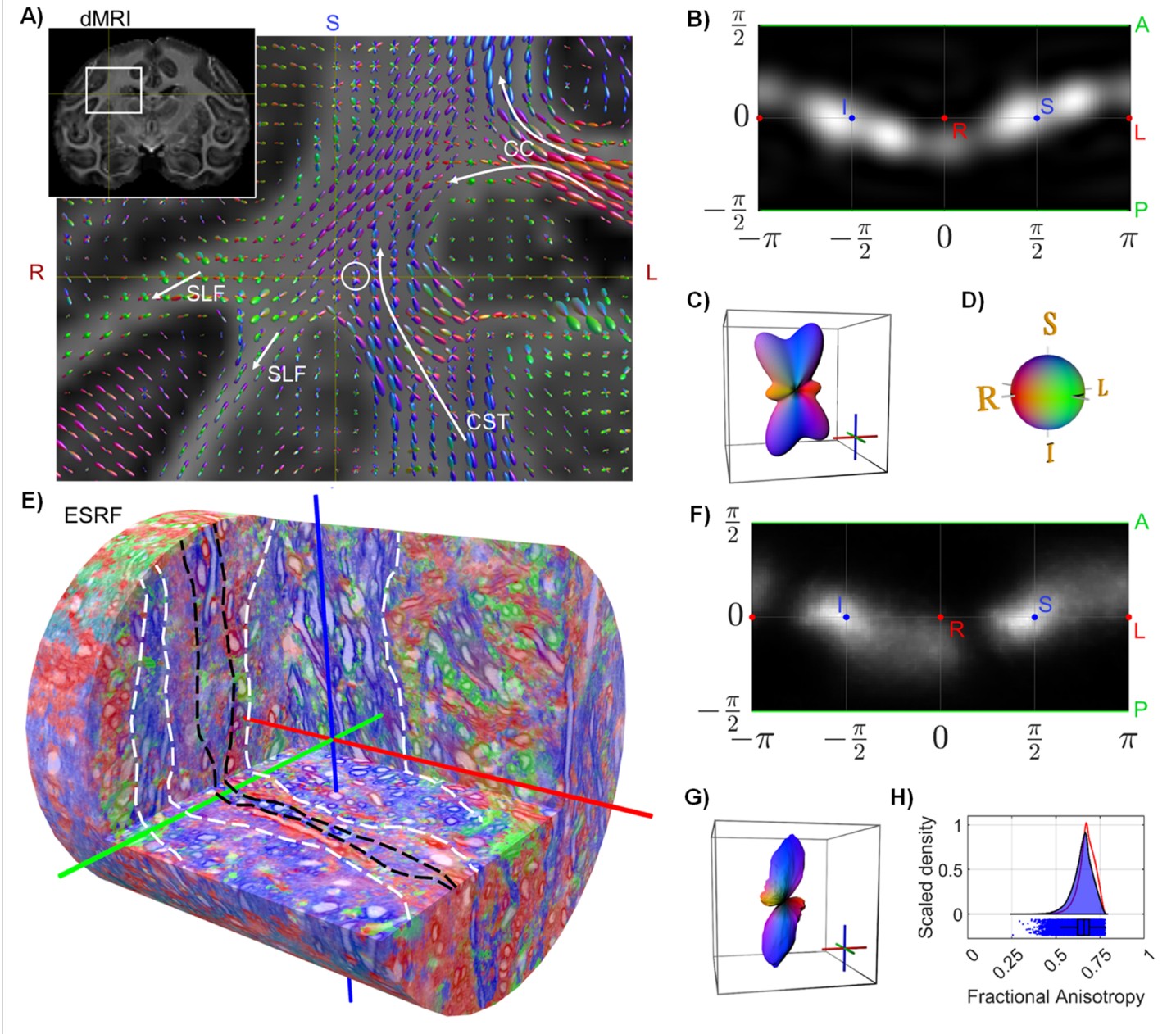

**Figure 4.** Results in complex monkey deep white matter (WM) region. **A**) Anatomy and constrained spherical deconvolution (CSD) glyphs are seen in the volume surrounding the local neighbourhood of the sampled biopsy location (white circle). (**B and C**) Results in the matching magnetic resonance imaging (MRI) voxel, showing (CSD) fibre orientation distribution (FOD) and the corresponding glyph. (**D**) The directional colour map used throughout the figure. (**E**) Example of rendering from an European Synchrotron (ESRF) volume with colouring corresponding to the structure tensor main direction. Marking with dashed lines indicates clear fasciculi. (**F and G**) Structure tensor directional statistics from the stacked field-of-view (FOV), showing the structure tensor FOD and corresponding glyph. (**H**) Structure tensor shape statistics from the stacked FOV, showing the kernel density estimate of fractional anisotropy (FA) values. The red curve is a copy of the FA distribution from the ESRF corpus callosum (CC) region (*Figure 3G*).

The online version of this article includes the following video for figure 4:

**Figure 4—video 1.** Animation related to *Figure 4E*.

https://elifesciences.org/articles/94917/figures#fig4video1

## Axon fasciculi trajectories in straight and complex tissue samples

To explore the macroscopic axonal organisation and trajectory variations of fasciculi at the MRI sub-voxel level, we applied streamlined tractography (*Tournier et al., 2019*) to the main direction of the structure tensor analysis of the synchrotron data. We manually drew seeding regions to delineate where we expected axons to project towards within the sampled volume (see Methods section). To ease visualisation and quantification, we used QuickBundle clustering (*Garyfallidis et al., 2012*) to group neighbouring streamlines with similar trajectories into a centroid streamline. This centroid streamline serves as an approximation of the actual trajectory of a fasciculus. For simplicity, we designate such a centroid streamline as streamline.

Tractography in the DESY corpus callosum data revealed evenly distributed streamlines throughout the sample (*Figure 5A*). The streamlines all follow a primary L-R direction, and a large portion of the trajectories follow a small S-I bending (coloured purple), which is in good agreement with the local FODs both from structure tensor and CSD dMRI data (*Figure 2* vs. *Figure 3*). Depicting the portion of streamlines without a strong S-I direction component with a separate colour (*Figure 5A*, cyan streamlines, and the supplementary animation, *Figure 5—video 1*) conveys the laminar organisation also observed in the structure tensor analysis (*Figure 3A*). The inclination angle between the trajectory of the two laminae is about 35 degrees.

For the ESRF data, the streamlines typically represent the trajectory of a small number of axons or, in some cases, of single large-diameter axons. As such, the streamlines in ESRF data (*Figure 5C*) better reflect detailed axonal trajectory variations as compared to the DESY data. For example, streamlines in the CC show non-straight trajectories that circumvent neighbourhood obstacles such as clusters of cell bodies, other axons, and blood vessels (orange structures). Interestingly, we observed a few axon fasciculi crossing the FOV orthogonally to the otherwise dominant R-L organisation. This phenomenon was particularly evident along the blood vessels, as illustrated in *Figure 5C* (red streamlines).

As expected, streamlines in the ESRF *centrum semiovale* data, followed the main direction of the laminae identified in the structure tensor directional map seen in *Figure 4F*. *Figure 5D* shows the trajectory of streamlines coloured according to the local direction, i.e., R-L (red), A-P (green) and S-I (blue). Independent of the main direction, we observe streamlines skirting around local obstacles and other axons, resulting in more complex and less straight trajectories than in the CC region.

To quantify the shape of fasciculi trajectories through the image volume, we characterised each streamline with two scalar values: (1) the streamline tortuosity index, which is a unitless score in [1, ∞] describing the deviation from linearity, and (2) the calculated maximum physical deviation from a straight line of each streamline, which depicts the maximal axonal 'amplitude' or dispersion. Both scalar metrics are independent of the general fasciculus orientation, but reflect micro-dispersion that can impact dMRI estimations (*Nilsson et al., 2012*; *Andersson et al., 2020*). We estimated the distributions of tortuosity and maximum deviation for all streamlines in both the DESY and ESRF samples (*Figure 5B*).

In the CC, the median tortuosity index of the distribution was almost identical in the DESY (1.01, IQR: 0.01) and ESRF sample (1.01, IQR: 0.01). A tortuosity index close to unity is equivalent to highly straight fibres, consistent with both the visual inspection of the streamlines (*Figure 5A and C*), the estimated ODI, and likewise with our expectation for the CC. In contrast, the median observed maximum deviation within the samples differed distinctly between DESY (17.1 μm, IQR: 10.7 μm) and ESRF (2.8 μm, IQR: 3.1 μm). In ESRF data, the small FOV limits the analysis to capture only microscopic effects. Therefore, the maximum deviation is driven by the interaction of the fasciculus with neighbourhood obstacles, i.e., micro-dispersion. We observed maximum deviations as high as 12 μm, which were due to obstacles like blood vessels, thus in good agreement with observations reported in *Andersson et al., 2020*. In DESY data, deviations were as high as 40 μm, which we attribute to more macroscopic effects such as the bending of pathways, as observed in *Figure 5A*. We suppose that this may arise from the U-shaped macroscopic trajectories of CC fibres towards the cortex.

In the ESRF *centrum semiovale* sample, the median tortuosity index was significantly higher and more variable (1.04, IQR: 0.07) than in the CC. This agrees with the more complex crossing fibre trajectories observed in *Figure 5D* (and the supplementary animation, *Figure 5—video 2*). Interestingly, the distribution of maximum deviations had a peak of around 5.3 μm and a median of 7.1 μm (IQR: 9.6 μm). While it is an increase compared to the CC, the magnitude of streamline dispersion is on a comparable scale.

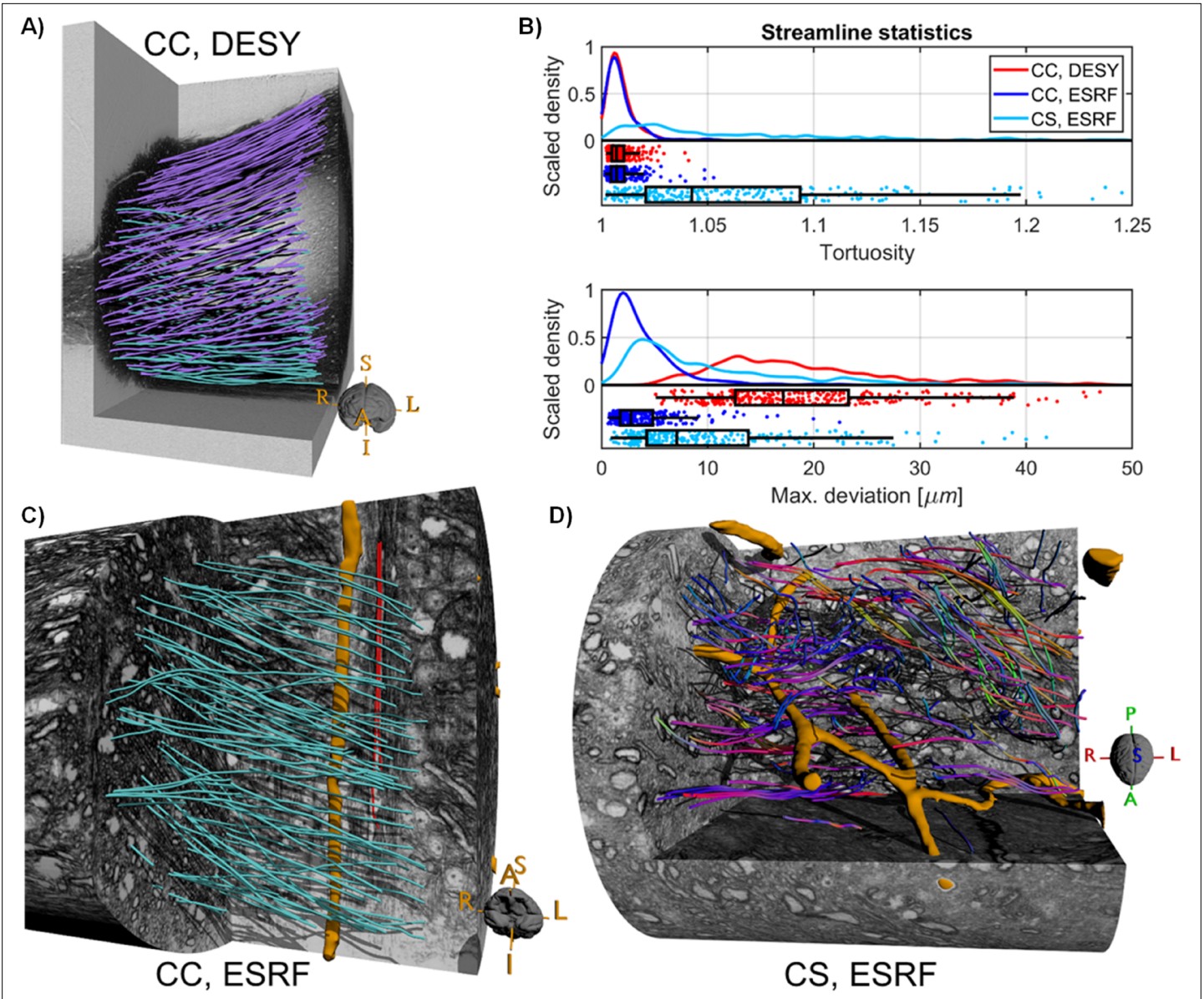

**Figure 5.** Structure tensor-based tractography for the monkey samples. (**A**) Streamlines in the corpus callosum (CC) sample from Deutsches Elektronen-Synchrotron (DESY). Purple streamlines have a strong upward directional component, unlike the cyan streamlines (assuming that streamlines travel from right to left). (**B**) The statistics of streamlines quantification using tortuosity index and maximum deviation. (**C**) A selection of streamlined in the CC within a single ESRF scan. Cyan streamlines have R-L as the strongest directional component, whereas red streamlines do not. The orange structure is a segmented blood vessel. (**D**) Streamlines in the complex *centrum semiovale* region (CS) within a single ESRF scan. Streamlines are coloured according to their local main direction. Orange structures represent segmented blood vessels.

The online version of this article includes the following video(s) for figure 5:

**Figure 5—video 1.** Animation related to *Figure 5A*.

https://elifesciences.org/articles/94917/figures#fig5video1

**Figure 5—video 2.** Animation related to *Figure 5D*.

https://elifesciences.org/articles/94917/figures#fig5video2

We conducted three non-parametric statistical tests of significance (α=0.05) between the two CC samples (DESY vs. ESRF) and between the two ESRF samples (CC vs. CS). These tests included: a Kolmogorov-Smirnov (two-sample) test for equality of distributions, a Wilcoxon rank sum test for equality of medians, and a Brown-Forsythe test for equality of variance (*Hollander et al., 2015*; *Brown and Forsythe, 1974*; *Massey, 1951*). For streamline tortuosity (*Figure 5B*, top), the null hypothesis

**Table 1.** Data samples included in this study.

All voxel sizes are isotropic. The field-of-views (FOVs) of ID-2 and ID-3 are given as four stitched scans (marked with *). Individual FOVs were 0.21×0.21×0.21 mm.

| ID | Specimen | Synchrotron | Sample | FOV [mm] | Voxel size [nm] | |
|---|---|---|---|---|---|---|
| 1 | Monkey | DESY | Corpus callosum (CC), midbody | 0.54×0.99 ×0.99 | 550 | |
| 2 | | ESRF | | 0.78×0.21 ×0.21* | 100 | — |
| 3 | | ESRF | Centrum semiovale (CS) | 0.78×0.21 ×0.21* | 100 | |
| 4 | Mouse | DESY | Corpus callosum, splenium & cingulum | 0.63×0.35 ×0.35 | 550 | |
| 5 | | ESRF | | 0.24×0.24 ×0.24 | 75 | — |
| 6 | CPZ Mouse | DESY | | 0.64×0.38 x 0.50 | 550 | |
| 7 | | ESRF | | 0.30×0.30 ×0.30 | 100 | |

was accepted for all three tests between CC samples (p=0.2, p=0.7, p=0.2), but rejected for all tests comparing CC and CS (p<0.001). For streamline maximum deviation (*Figure 5B*, bottom), all six null hypotheses were rejected (p<<0.001).

## Multi-scale, multi-modal imaging data in healthy and demyelination mouse brains

Finally, we investigated the organisation of fasciculi in both healthy mouse brains and a murine model of focal demyelination induced by 5 wk of cuprizone (CPZ) treatment. This allowed for the exploration of the disease-related influence on axonal organisation, particularly under inflammation-like conditions with high glial cell density at the demyelination site (*He et al., 2021*). The experimental setup for DESY and ESRF is similar to that described for the monkey, with the exception that we did not perform dMRI and synchrotron imaging on the same brains, and only collected MRI data for healthy mouse brains. This approach allowed us to apply the same structure tensor and tractography streamline analysis used previously, but in a healthy versus disease comparison, demonstrating the methodology's ability to provide insights into pathological conditions.

The dMRI is a whole-brain FOV, whereas the FOVs of DESY and ESRF are similar to those for the monkey (see *Table 1*). The image voxels of dMRI, DESY and ESRF have isotropic side lengths of 125 μm, 550 nm and 75–100 nm, respectively. Although the mouse brain image resolution of the DESY and ESRF data is similar to that acquired on the monkey, the white matter structures in the mouse brain are considerably smaller in proportion, and the resolvable anatomical length scale differs. In the mouse, the FOV of the DESY data included both the splenium region of the corpus callosum and parts of the neighbouring Cingulum pathway (*Figure 6A*). Furthermore, in the ESRF data, the image resolution did not suffice to outline the diameters of even the largest axons (*Figure 6B*), in contrast to the monkey CC. However, blood vessels and glial cells are outlined as the stained myelinated axons skirt around them. Unlike in the monkey samples, we did not observe vacuoles in the mouse samples.

DESY and ESRF data both revealed demyelinated regions in the CC of the cuprizone-treated mouse. The lesion area is demarcated by an intensity gradient increasing from dark (intact myelination) toward a brighter signal (demyelination) (*Figure 6C and D*). In the demyelinated area, the ESRF data revealed a higher density of cell bodies and small accumulations of heavily stained material. Cellular structures were manually segmented in a small representative region to reveal their 3D shapes (*Figure 6E*).

## Preservation of the organisation of major pathways in healthy and demyelination mouse brains

In dMRI of the healthy mouse brain, the colour-coded diffusion tensor directions of the first (major) principal direction have the same R-L direction (red) as seen in the monkey (*Figure 7A* vs. *Figure 2A*). However, *Figure 7A* shows a shift of the second and third principal directions in the mouse CC, as compared to the monkey.

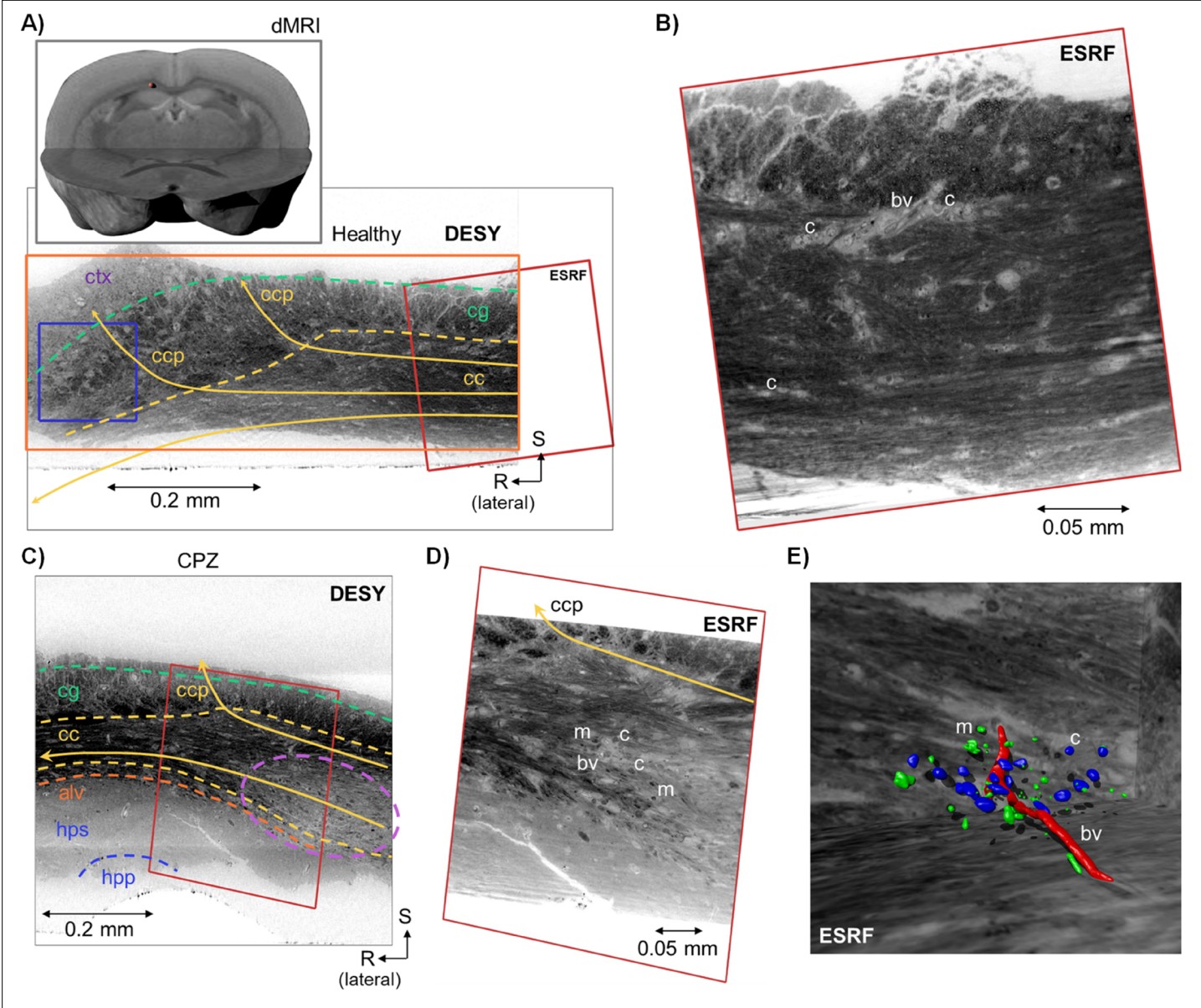

**Figure 6.** Overview of the mouse datasets. (**A**) Location of the sample within the mouse brain and a coronal slice from the Deutsches Elektronen-Synchrotron (DESY) dataset, with indications of anatomic regions: cc = corpus callosum, cg = cingulum, ctx = cortex, and ccp = cortical projections. The blue box indicates the size of a DWI voxel. (**B**) The co-registered slice from the European Synchrotron (ESRF) volume (position indicated by the red frame). Labels indicating blood vessels (bv) and cells (**c**), (**C**) Coronal slice from the DESY dataset of a cuprizone (CPZ)-treated mouse, with indication of anatomic regions: cc = corpus callosum, cg = cingulum, alv = alveus, hps = hippocampal striatum, hpp = hippocampal pyramidal layer, ccp = cortical projections. The dashed purple line shows a demyelinated region of the CC. (**D**) The co-registered slice from the ESRF volume (indicated by the red frame). Labels indicate blood vessels (bv), cells (**c**), and myelin 'debris'/macrophages (**m**). (**E**) 3D rendering of a local segmentation in the corresponding region in **D**. The blood vessel is coloured in red, cells in blue, and 'myelin debris' in green. This segmentation was done manually using ITK-SNAP (RRID:SCR_002010).

In the healthy and demyelination mouse brains, we use only a single scale-space level for the structure tensor patch size (see method section) both in the DESY and ESRF data sets. The integration patch sizes correspond to 18.7 μm for the DESY and 9.4 μm for the ESRF samples (see *Table 2*).

In the DESY data from healthy mice, the directional colour coding of the structure tensor was in accord with the diffusion tensor MRI (*Figure 7A* vs. *Figure 7B*). We found sporadic directional deviations in the coronal view, represented as local regions with different directional colours in *Figure 7B*, which indicate local crossing axon fasciculi and/or blood vessels. Nevertheless, the visualisation of

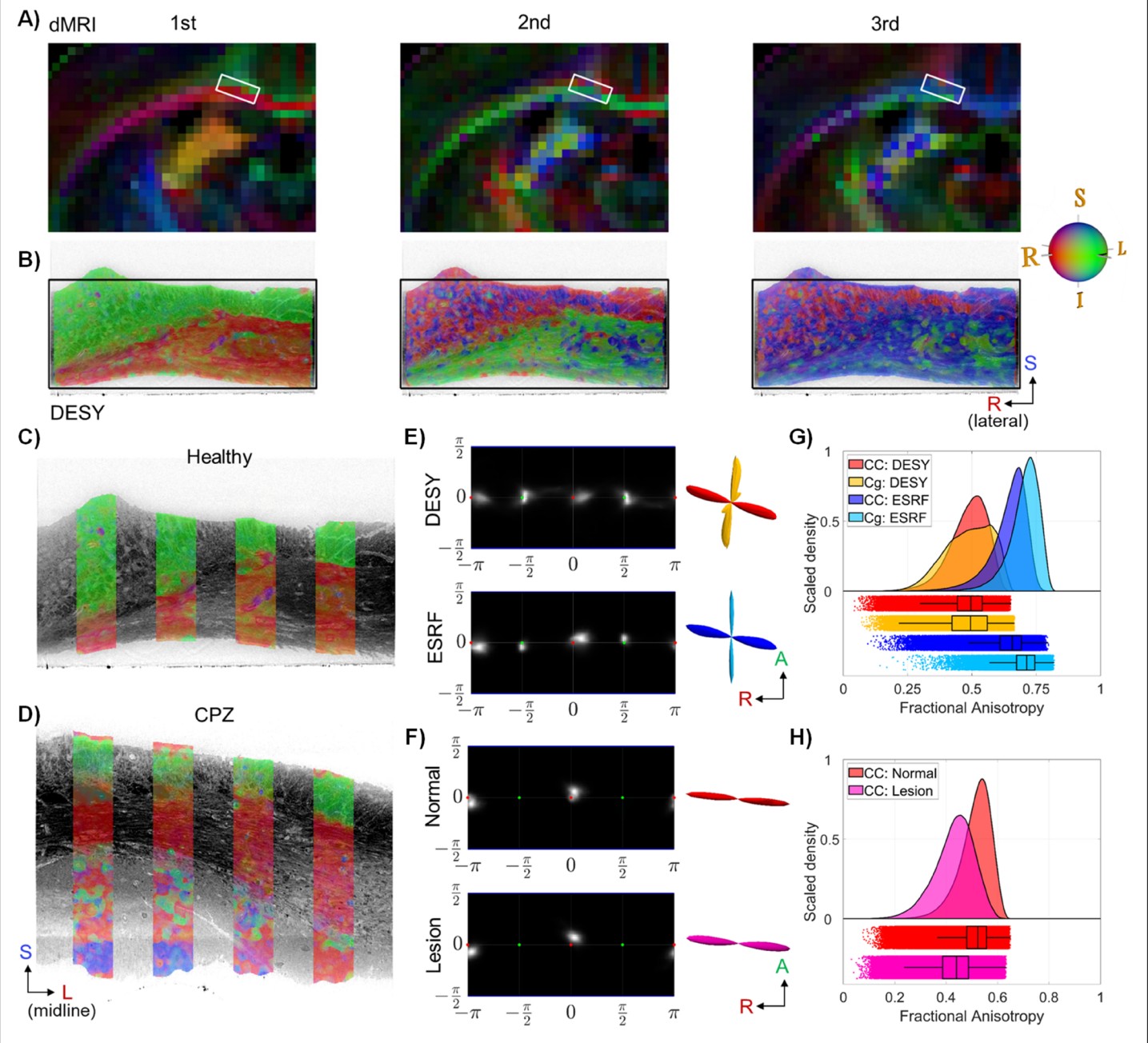

**Figure 7.** Mouse diffusion tensor and structure tensor results. (**A**) and (**B**): The directional components of the diffusion tensor and the structure tensor in a coronal slice from a healthy mouse brain. The white frame in (**A**) indicates the approximate size and location of the Deutsches Elektronen-Synchrotron (DESY) field-of-view (FOV), i.e., the black frame in (**B**). (**C**) and (**D**) The main structure tensor direction from the DESY data overlaid on a slice from a healthy mouse brain (**C**) and a cuprizone (CPZ)-treated mouse (**D**). (**E**) Structure tensor fibre orientation distributions (FODs) from a healthy mouse along with corresponding glyphs. The glyph colouring indicates whether the FOD contribution is from the corpus callosum (CC) (red/blue) or cingulum (yellow/cyan). (**F**) structure tensor FODs from the CPZ-treated mouse CC (DESY), split into contributions from a normal appearing region and a demyelinated area. (**G**) and (**H**) structure tensor fractional anisotropy (FA) distributions from various regions of healthy and lesioned mouse samples.

orientations did not reveal any axonal organisation in the mouse CC due to the lack of local angular contrast, unlike the clear laminar structures seen in the monkey sample (*Figure 3A*). Any parallel organisation in tissue remains undetectable because our visual contrast relies on angular differences.

The two major pathways, the CC and cingulum, appear in the mouse as two sharply separated structures, facilitating the generation of image masks for regionally independent quantifications of the structure tensor analysis. These structures cross almost orthogonally, as shown in the 3D glyphs of the

**Table 2.** Structure tensor parameter values used for the samples in this study (see Table 1 for sample ID).

The kernel size represents the width of the ρ-kernel converted to physical distance in accordance with the voxel size of the specific dataset. The scale-space approach was applied only to the monkey ESRF samples, with the range of scales listed in *Table 2*. In all other cases, the image resolution was too low compared to the anatomical scale for structures to visually present with distinguishably different sizes. Therefore, there was no benefit in the scale-space approach. Instead, the standard ST analysis with a fixed kernel size was used.

| ID | Voxel size [nm] | Scaling factor | ST-parameters ($\rho,\sigma$) [voxels] | | ST conversion ($\gamma$) | Kernel size [μm] |
|----|-----------------|----------------|----------------------------------------|----------|--------------------------|------------------|
| 1 | 550 | 2 | 2.5 | 0.5 | 0.30 | 12 |
| 2 | 100 | 4 | 8 scales* | 8 scales* | 0.30 | [9.2–2] |
| 3 | 100 | 4 | 8 scales* | 8 scales* | 0.30 | [9.2–2] |
| 4 | 550 | 2 | 4 | 1 | 0.35 | 18.7 |
| 5 | 75 | 5 | 6 | 1 | 0.25 | 9.4 |
| 6 | 550 | 2 | 4 | 1 | 0.35 | 18.7 |

*Scale space structure tensor parameters: $\rho$ = [5.50, 4.50, 3.50, 3.50, 2.50, 2.50, 1.50, 1.00], σ = [3.00, 2.75, 2.50, 1.50, 1.50, 1.00, 1.00, 0.50].

structure tensor FODs from both the ESRF and DESY datasets (*Figure 7E*). In all cases, the ODI was estimated to be below 0.06, indicating highly directed pathways, similar to those values measured in the monkey CC. The DA indices show more variation: In the CC, DA was 0.49 and 0.32 for the DESY and ESRF samples, respectively, and in the cingulum, 0.78 and 0.29. These values are on the same scale as those for the monkey and are non-zero, suggesting some axonal organisation is present, even though it was not visually confirmed.

The structure tensor FA histograms of the mouse CC and cingulum, are both shifted toward lower values in the DESY data as compared to the ESRF data (*Figure 7G*). The two histograms overlap in the DESY data, whereas the ESRF data shows a shift towards higher FA values in the cingulum as compared to CC, indicating a dependency of the estimated FA on image resolution relative to the size of structures. From the ESRF data, we were able to roughly segment cell clusters and blood vessels. Since the volume consists of about 13.5% cells/blood vessels in the cingulum, versus only 8.5% in CC, the higher density of extra-axonal structures may factor into the differences in estimated FA values.

In the CPZ-treated mouse brain, the directional colour-coded structure tensors in the DESY data match that in the healthy mouse, even in the demyelination region, which shows reduced image contrast (*Figure 7C vs. D*). The fact that there even is directional contrast in the demyelinated region, depicts the advantage of phase-contrast imaging, where the obtained contrast is not solely dependent on absorption. Note that the demyelination region is confined to axons within the CC, with sparing of the cingulum. The lower image contrast (lack of strong edges) in the demyelinated region contributes to the blurrier tensor shapes and lower FA values as compared to the normal-appearing white matter (mean FA: 0.44 vs 0.52), as shown in *Figure 7H*. The FODs are similar (*Figure 7F*) with ODI = 0.03 and 0.04 for the normal-appearing and lesioned regions respectively as expected for the highly aligned axons in CC. In the same order DA = 0.18 and 0.40.

## Axon fasciculi trajectories in healthy and demyelination mouse brains

For better comparisons, we registered the ESRF data of mouse CC onto the DESY data. In healthy and cuprizone-lesioned mice, *Figure 8A and B* shows that the CC and cingulum are both mappable using streamlines. Irrespective of image resolution, the cingulum did have a lower tortuosity and maximal deviation than the CC (*Figure 8C and E*), which could be due to differences in densities of extra-axonal structures. This is suggested by the streamlines projecting through the demyelinated region (*Figure 8B*), which contains more cells (*Figure 6D and E*). Indeed, both indices broadened with lower image resolution in combination with a larger FOV that covers macroscopic shape changes. An example is the bending of the corpus callosum detected by the longer streamlines.

Like before, we conducted three non-parametric statistical tests of significance (α=0.05): a Kolmogorov-Smirnov test, a Wilcoxon rank-sum test, and a Brown-Forsythe test. These tests compared measurements between the two pathways from each dataset (CC vs. Cg) and within the same pathway

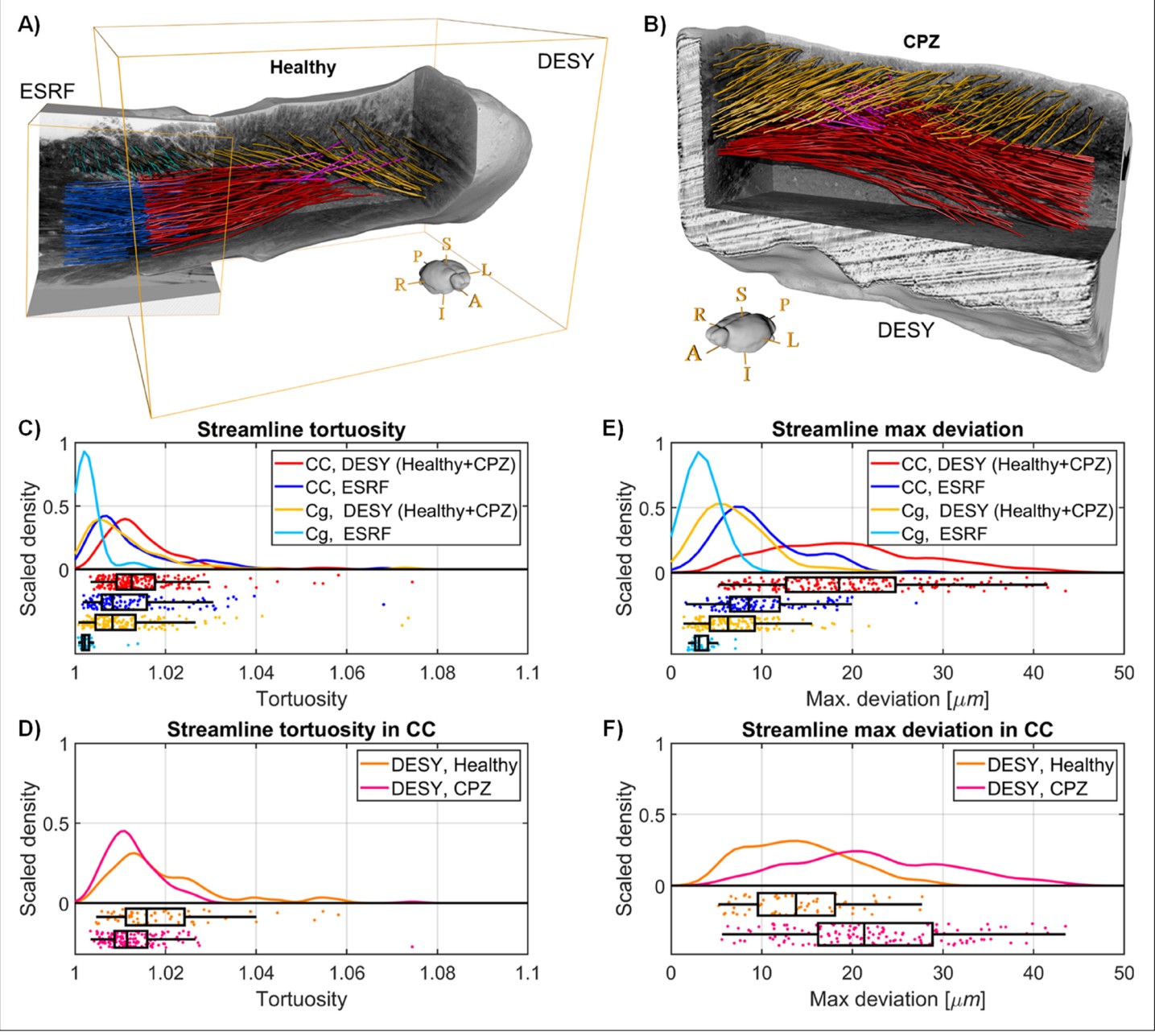

**Figure 8.** Mouse tractography results. (**A**) Tractography visualisation from a healthy mouse brain sample. Red and Blue: Streamlines in corpus callosum (CC). Yellow and Cyan: Streamlines in cingulum. Pink streamlines are extrapolations of CC streamlines that represent plausible cortical projections through the cingulum. (**B**) Tractography visualisation from Deutsches Elektronen-Synchrotron (DESY) data of a cuprizone (CPZ)-treated mouse brain. (**C**) and (**E**) Tractography streamlines centroid statistics from the healthy mouse brain synchrotron volumes. The curves are kernel density estimates of the tortuosity and maximum deviation, respectively. (**D**) and (**F**) Statistics for the streamlines in the corpus callosum of the DESY healthy and CPZ mouse brain samples.

across different modalities (DESY vs. ESRF). For tortuosity (*Figure 8C*), the null hypotheses for distribution and median equality were rejected in all cases (p<<0.001). The null hypotheses for variance equality were accepted when comparing the CC across samples (p=0.35, red vs. blue) and comparing the CC and cingulum in the DESY dataset (p=0.66, red vs. yellow), but were otherwise rejected (p<<0.001). For maximum deviation (*Figure 8E*), all 12 null hypotheses were rejected (p<<0.001).

Interestingly, both in healthy and demyelination samples, some streamlines from the CC corpus deviate to project into the cortex, as illustrated in *Figure 6* and *Figure 8A and B* (pink extensions of

streamlines). Those that project into the cortex also tend to originate from a more superiorly placed seed point. As such, the streamlines visually present a vertically layered organisation within the CC.

## Discussion

By combining diffusion MRI and high-resolution phase-contrast synchrotron imaging, we quantified the 3D organisation of fibre pathways in white matter across a range of anatomical length scales. Importantly, our findings revealed common principles of fibre organisation in both monkeys and mice despite differences between species; small axonal fasciculi and major bundles formed sheet-like laminar structures, with relative angles that depended on the characteristics of the major pathways to which they belonged. By applying a scale-space structure tensor analysis and streamline tractography to the synchrotron datasets, we quantified the micro-dispersion of individual axons and fasciculi. Interestingly, the dispersion magnitude is indicative of fasciculi that skirt around obstacles in the white matter such as cells and blood vessels, and the results are observed across both white matter complexity (straight vs crossing fibre region) and pathology. Our dMRI and scale-space structure tensor analysis allowed us to compare and quantify tissue anisotropies and fibre orientations. We found that FODs were comparable across image resolutions and modalities. The observed discrepancies can be attributed to the fact that the FOVs are not exactly matched. Estimates of structure tensor-derived microscopic FA show a clear pattern across modalities. However, to achieve good specificity in the anatomical feature correlation the diffusion tensor and structure tensor models must be sensitive to the same anatomical length scale.

### Laminar organisation across image resolutions and modalities

Our white matter samples from monkey and mouse brains were geometrically organised as stacked laminae across anatomical length scales extending from individual axons to axon fasciculi and the major WM pathways.

In the monkey CC DESY data, which has a field of view (FOV) comparable to a dMRI voxel, a columnar laminar organisation at a macroscopic level was visually revealed from the structure tensor (ST) direction colouring. However, this laminar organisation was not visible in the higher-resolution ESRF data for the same tissue sample. Although the two samples were not co-registered, the size of a single ESRF FOV within the DESY sample is illustrated in *Figure 3A*. This demonstrates the possibility of placing the ESRF sample where the observed laminar structure is absent. Consequently, knowledge of the tissue structural organisation and its orientation is important to fully benefit from the stacked FOV of the ESRF sample and when choosing appropriate minimal FOV sizes in future experiments.

Interestingly, when characterising FODs with measures like ODI and DA as indicators of fibre organisation, rather than relying on visualisation, results from large- and small-FOV data show no discrepancies. This statistical approach discards the spatial context (visually perceived as laminae), highlighting the need to combine both methods.

Invasive tracer studies and histology in monkeys (*Caminiti et al., 2009*; *Howard et al., 2023*) and histology of humans (*Mollink et al., 2017*) previously hinted at a columnar organisation in the ventral part of midsagittal CC. However, the earlier approaches failed to show the finer 3D columnar organisation of 10 µm thick laminae as revealed by the present structure tensor analysis of the synchrotron data. The occurrence of these thin layers with slightly different angles may be confirmed by visual inspection of polarised light imaging (PLI) data collected in a human brain coronal CC slice (in Figure 6 of *Mollink et al., 2017*). In a coronal view of the mid-sagittal region, PLI showed fibres arising from the right and left hemispheres reaching the midsagittal CC as a layered structure, based on its orientational colour coding. The greater slice thickness (in PLI 100 µm) in a coronal view may have resulted in a partial volume effect arising from the thinner non-parallel columnar laminar sheets that we observed in the sagittal view of our synchrotron data (*Figure 3A*).

Structure tensor analysis of the present ESRF data in the complex fibre region clearly shows the crossing of three pathways composed of laminar sheets of fasciculi (*Figure 4E*). Compared to the parallel within-tract axonal fasciculi in the monkey CC, the boundaries between fasciculi in the complex fibre region are thin (tens of micrometres), and, therefore, likely to be less well defined when visualised as streamlines. However, we did not see such an intermingling of streamlines at the

interface between the crossing of the larger well-defined tracts such as the CC and cingulum in the mouse brain (*Figure 7B–D*).

## The inclination angle between fasciculi

The structure tensor method performed very well in analysing the fasciculi without segmentation, through its detection of tracts and fasciculi based on their angular differences and highlighting of their sheet-like and laminar organisation. The observed tendencies regarding fasciculi organisation can be grouped into two: Within-WM pathways where inclination angles are low, including 0 degree parallel alignment, and between-WM pathways where the angles are high.

Within major WM pathways, neighbouring axonal fasciculi appeared to follow the same principal direction but could have inclination angles up to the order of 35 degrees in both species.

In the monkey CC (mid-body), we observed laminar organisation indicated by clear spatial angular differences in the ST directions in the sample (*Figure 3A*). Quantifications of the FOD shape showed DA indices of 0.55 and 0.59 for the DESY and ESRF samples, respectively. In contrast, the mouse CC (splenium) did not visually reveal a similar angled laminar organisation (*Figure 7C*), and the DA indices were lower, at 0.49 and 0.32, respectively. Two possible explanations exist. First, the within-pathway laminar organisation may not be identical across the entire CC. Consequently, more scans from other CC regions would be required to confirm. Second, the different species might account for the differences. Larger brains like the monkeys might foster a different level of within-pathway axon organisation compared to the smaller mouse. Although we could not visually detect laminar organisation from the colour coding of the ST direction in the mouse, the non-zero DA values suggest some level of organisation. This is supported by our streamline tractography, which indicates a vertical layered organisation (*Figure 8A and B*). It further aligns with studies using histological tracer mapping that shows a stacked parallel organisation of callosal projections in mice, between cortex regions M1 and S1 (*Zhou et al., 2013*). Nevertheless, we cannot rely solely on voxel-wise ST directions to fully describe the axonal organisation, as this method does not contrast almost parallel fasciculi (inclination angles approaching 0 degrees). Analysing patterns in tractography streamlines would be an interesting future direction for this purpose.

Interestingly, the default multi-fibre dMRI CSD model for the monkey CC showed only a single fibre direction with an isotropic FOD, indicating a coherent fibre pathway (*Figure 2C*). This approach did not reveal any of the layering and organisation evident from structure tensor analysis of the synchrotron data (*Figure 3*). Nevertheless, when investigating the directions of the diffusion tensor model we see signs that the diffusion signal is indeed sensitive to the complexity of fasciculi organisation; the principal direction agrees with the CSD and is locally consistent across the entire CC. However, the second and third eigenvectors are also informative, in that they behave symmetrically across the midline, and are locally consistent, although their order can shift at specific locations. This is likely an indication of the local organisation, similar to PLI-based observations (*Mollink et al., 2017*). Additionally, *Tax et al., 2016* have demonstrated the calculation of a grid-crossing sheet probability index from diffusion MRI data, which suggested the presence of sheet-like features in a crossing fibre region, which is in line with our findings in the synchrotron data. Note that the method by Tax et al. only detects sheet-like structures crossing on a grid and does not reveal laminar structures with lower inclination angles, as we observed in the monkey CC.

By further exploring the diffusion tensor modelling of the CC in the Human Connectome Data set (N=4, *Appendix 1—figure 1*) we found an order of the second and third eigenvectors resembling that in the dMRI of the monkey. This observation supports the future exploration of the diffusion-weighted signal to reveal insights about axonal organisation within a WM pathway. For the present, we have demonstrated axonal organisation only in selected regions of the WM, but further research may establish more generalised rules governing fasciculi organisation in WM pathways, and potential species-related differences.

Between-WM pathways, the fasciculi cross with high inclination angles approaching 90 degrees. This is readily detectable in diffusion MRI multi-fibre CSD modelling (*Tournier et al., 2004*) and in the structure tensor analysis. We observed this right angle crossing at the *centrum semiovale*, a complex fibre region with the crossing of three different pathways (*Figure 4*). Using tracer labelling of axonal projections, *Mortazavi et al., 2018* revealed grid-like right-angle crossing of axons in a similar region in the monkey brain. Tractography and micro-dissection findings of *Wedeen et al., 2012* of the same region similarly showed grid-like crossing of laminar from different pathways.

Interestingly, we observed in the ESRF data a single axonal fasciculus composed of a few axons that crossed the main callosum pathway at an almost 90 degree angle running parallel to a blood vessel (*Figure 5C*). Such phenomena may be overlooked in lower resolution data and have such a small contribution to the diffusion MRI signal as to be dismissed as noise/artefact. Without prior knowledge of their existence, such crossings are apt to be disregarded by modelling choices, such as orders of spherical harmonics, model regularisation, and number of fibre populations.

Our findings of laminar organisation and the distinction between high and low inclination angles seem to support a simple topological rule across anatomical length scales. This holds especially for the low inclination cases, which retain their laminar organisation with minimal intermingling in the neighbourhood, meaning that we expect that topological organisation should be kept at a distance. If so, the (inclination angle) information might serve to form rules for low-resolution diffusion MRI-based tractography about how best to project through crossing fibre and bottleneck regions, which is currently a source of false-positives trajectories (*Maier-Hein et al., 2017*). The reason is that standard tractography methods do not 'remember' or follow anatomical organisation rules as they trace through complex regions. Our findings on pathway lamination and inclination angles—low for parallel-like trajectories and high for crossing-like trajectories—can help incorporate trajectory memory into these methods, reducing the risk of false trajectories.

We believe our observed topological rule of white matter laminar organisation can be explained by a biological principle known from studies of nervous tissue development. The first axons to reach their destination, guided by their growth cones, are known as 'pioneering' axons. 'Follower' axons use the shaft of the pioneering axon for guidance to efficiently reach the target region (*Breau and Trembleau, 2023*). Axons can form a fasciculus by fasciculating or defasciculating along their trajectory through a zippering or unzipping mechanism, controlled by chemical, mechanical, and geometrical parameters. Zippering 'glues' the axons together, while unzipping allows them to defasciculate at a low angle (*Šmít et al., 2017*). Although speculative, the zippering mechanism may be responsible for forming the laminar topology observed across length scales. The defasciculation effect can explain our results in the corpus callosum (CC) of monkeys, with laminar structures at low angles (~35 degrees) also observed by *Innocenti et al., 2019*; *Caminiti et al., 2009*, as well as in other major pathways (*Sarubbo et al., 2019*). In contrast, a fasciculation mechanism may be observed in the mouse CC (0 degrees). If the geometrical angle between two axons is high, i.e., toward 90 degrees, the zippering mechanism will not occur, and the two axons (fasciculi) will cross (*Šmít et al., 2017*). This supports our and other findings that crossing fasciculi or pathways occur at high angles toward 90 degrees in the fully matured brain (*Wedeen et al., 2012*). Once myelination begins, the zippering mechanism is lost (*Šmít et al., 2017*), suggesting that laminar topology is established at the earliest stages of brain maturation.

## Sources to the non-straight trajectories of axon fasciculi

Applying streamlined tractography to the main fibre orientations in the structure tensor analysis enabled a geometric quantification of the non-linearity of axon fasciculi trajectories as a tortuosity metric (*Pingel et al., 2022*). Surprisingly, we observed that the maximal amplitude of the axon fasciculi was comparable when measured in the crossing and straight fibre regions. This suggests that the geometrical variation in an axon fasciculus trajectory may hold similarly throughout the white matter.

We also detected differences in minor pathways: In the mouse brain ESRF data, axon fasciculi were straighter (i.e. having lower tortuosity) in the cingulum than those in the corpus callosum. Similarly in monkey ESRF data, fasciculi were straighter in the CC than in the crossing fibre region.

For the monkey and mouse brain samples, the distributions of maximal deviation in the ESRF and DESY data showed main peaks around 5–7 μm. This range of maximal deviation aligns with the typical radius of the cell bodies. Indeed, axons and fasciculi have been observed to skirt around oligodendrocytes in the ESRF monkey data set (*Andersson et al., 2020*). In addition, the maximal deviation is in accord with the size of a single axonal fasciculi (*Figure 5C*) or blood vessels.

Even in the case of regional demyelination in the cuprizone-treated mice, we saw minor changes in tortuosity compared to normal axon fasciculi (*Figure 8D*). We suppose that microglia and macrophages invading the site of demyelination are sufficiently similar in size to oligodendrocytes that they have little net effect on the distribution of tortuosity. The measured max deviation was higher in the cuprizone-treated mouse (*Figure 8F*), but given the magnitude of approx. 30 μm, this effect is more

likely a description of the macroscopic bending of the CC, rather than a result of the demyelination. As our streamlines went through both demyelinated and normal-appearing regions of the CC, we experienced a similar dilemma found in MRI-based tractography applied in humans. The streamlines are based on only the direction of the tensor and are not guaranteed to be sensitive to local pathology. Additionally, demyelination is a dynamic process. We only sampled a single time point, and more must be included to explore if the tractography-based analysis has the sensitivity to correlate with the temporal dimension of demyelination.

Diffusion MRI is sensitive to the micro-dispersion effects (*Lee et al., 2019*; *Andersson et al., 2020*), but our findings suggest that the amplitude of micro-dispersion is primarily related to the sizes of extra-axonal structures independent of the complexity of the major white matter pathway. We also observe that the tortuosity can change between different white matter pathways. Assuming that the local distributions of cell bodies, vessels, and axon fasciculi can generalise within a pathway, then we can expect the micro-dispersion effects on the dMRI signal to be homogeneous.

Notably, high-resolution light-microscopy-techniques such as Polarised Light imaging with in-plane 5 µm and 30–100 µm thick slices (*Axer et al., 2011*) as well as light-sheet imaging are expected to be too low in resolution to detect the micro-dispersion effects.

## Fibre orientation distributions across image resolutions depend on FOV

We found a general agreement in the FODs between the different image modalities and across image resolutions. This is in line with other studies comparing dMRI with structure tensor analysis using both single and multi-fibre models (*Khan et al., 2015*; *Budde and Frank, 2012*; *Leuze et al., 2021*; *Schilling et al., 2016*).

The FOD discrepancies that we did observe could be attributed to the differences in the FOVs across the imaging modalities. As shown in *Figure 1*, we prepared biopsy samples for synchrotron imaging that was on a sufficiently large scale to cover several MRI voxels of isotropic 500 µm size. Indeed, the DESY sample covered multiple MRI voxels, except that a relatively large unstained region in the middle of the sample was not included in the generated FOD. Similarly, the four stacked image volumes of the ESRF measuring 780 µm in length by 210 µm in diameter only covered a fraction of the DESY volume and MRI image voxel. Since the FOD is dependent upon fibre organisation, the observed FOD differences reflect actual anatomical differences, for example with more volumetric weighing in one fibre direction. Therefore, although FODs are independent of image resolution, care must be taken when comparing two modalities without covering the same 3D volume, as observed in the CC (*Figure 3E and F* vs. *Figure 3H, I*). Examples are validation studies comparing the FOD from diffusion MRI fibre models with that derived from 2D or 3D histology which only rarely cover the same identical volume (*Khan et al., 2015*; *Budde and Frank, 2012*). In cases of differing FOV, there may be a risk of misattributing an actual difference in anatomical information across length scales as a methodological difference.

## Tissue micro anisotropy across modalities, image resolutions, and pathology

We show that tissue anisotropy metrics are comparable across modalities but vary with the anatomical length scale. The simple FA metric in the diffusion tensor MRI model was, as expected, sensitive to the average tissue organisation/anisotropy in the voxel. Thus, we found higher anisotropy in the CC (*Figure 2C*) as compared to the *centrum semiovale*, a more complex fibre region, much as observed by others (*Andersen et al., 2020*). In contrast, the µFA dMRI metric (*Kaden et al., 2016*; *Jespersen et al., 2013*; *Jespersen et al., 2013*; *Lasič et al., 2014*) is sensitive to tissue anisotropy only in microdomains of 10 µm length scale (molecular displacement). The µFA generates a mean anisotropy value within a voxel, returning similarly high FA values in both the CC and *centrum semiovale*, thereby confirming the expected independence of FA on the structural organisation (*Andersen et al., 2020*).

In the DESY and ESRF data, anisotropy measured by the structure tensor model is also determined within a micro-domain controlled by the patch-size and produces a distribution of values from the full volume. We introduced a scale-space parameter that automatically adjusts the patch-size to ensure optimal sensitivity to anisotropic features. Patch-sizes of 18 µm for DESY and 5 µm for ESRF data depict similarly sized micro-domains as in the µFA diffusion MR model and were shown to be largely independent of an axonal organisation (*Figure 4H* and *Figure 7G*).

The same anatomical features, namely cell membranes and myelin, dominate the contrast of diffusion vs. structure tensor techniques. In dMRI, these structures are the main sources for restricted and hindered water diffusion, and their osmium staining for the synchrotron scans provides strong image gradients modelled by the structure tensor. However, tissue preparation differs substantially for the two image modalities. In the case of ex vivo MRI, *Sun et al., 2005* showed that anisotropy of perfusion fixated, hydrated tissue should closely match that in vivo. However, synchrotron imaging calls for an extra dehydration step before tissue embedding in EPON, which changes the intra- and extracellular volume fractions due to tissue shrinkage (*Korogod et al., 2015*). Similarly, there are shrinkage effects between dehydrated and hydrated synchrotron image samples (*Töpperwien et al., 2019*). Given the large differences in tissue processing, the obtained anisotropy measures naturally differ across the modalities. Nevertheless, as both the diffusion- and structure tensor models were sensitive to the same anatomical features, the observed strong correlation in FA values was expected.

Across image resolutions, the distributions of structure tensor anisotropy also changed, being generally lower and broader in DESY data compared to ESRF data. This could reflect greater partial volume effects in the lower-resolution DESY data, thus decreasing the apparent separation of cells and axons. Such image blurring can change the image gradient information used by the structure tensor model, thereby making the anisotropy metric dependent on the image resolution. Therefore, structure tensor quantification of anisotropy metrics calls for invariant image resolution.

In our comparison of structure tensor anisotropy in healthy mice versus cuprizone-induced demyelination, the data clearly shows a lowered anisotropy distribution in focally demyelinated regions compared with healthy WM regions (*Figure 7H*). We observed an increased extra-axonal content in the lesion area, thus in agreement with the observed higher density of cells in demyelinated tissue (*He et al., 2021*). By outlining the various cellular structures in 3D, we could visualise what we believe to be cell bodies, macrophages, and the myelin debris engulfed inside macrophages (*Figure 6E*).

Interestingly, despite reduced anisotropy in demyelinated regions, the structure tensor still detects a clear directionally dependent anisotropy as in the healthy brain, consistent with the persistence of axons in this animal model despite demyelination (*He et al., 2021*). We did not collect diffusion MRI for the cuprizone-lesioned brains. However, we recently reported relatively preserved µFA values in demyelinated regions of the rat CC (*He et al., 2021*). Similar findings were made in a group of multiple sclerosis patients compared to normal (*Andersen et al., 2020*) supporting our synchrotron anisotropy results.

## Limitations

### Sample size

Increasing the number of samples across both species and examining laminar organisation at various length scales in more regions would strengthen our findings. However, securing beamtime at two different synchrotron facilities to scan the same sample with varying image resolutions is a limiting factor. Beamline development for multi-resolution experimental setups, along with faster acquisition methods, is a rapidly advancing field. For instance, the Hierarchical Phase-Contrast Tomography (HiP-CT) imaging beamline at ID-18 at the ESRF, enables multi-resolution imaging within a single session to address this challenge, though it is currently limited to a resolution of 2.5 µm (*Walsh et al., 2021*).

### Registration

It was not possible to realise perfect correlational cell-to-cell imaging between the three very different modalities and experimental set-ups. This resulted in minor cross-modal differences between FODs, for example in *Figure 3*, where we see a FOD tilt difference due to either misalignment or an anatomical difference, albeit without affecting the interpretation of quantitative measures.

MRI-to-synchrotron registration is challenging due both to the large resolution difference and different contrast mechanisms. Our best option was, therefore, to rely on prior knowledge of fibre orientations to match approximately the samples. Given that we know the site of biopsy sampling, we could confine our search to a small neighbourhood of possible/potential MRI voxels and identify the one giving the best visual match of fibre orientation.

The matching between the different synchrotron volumes was easier and achievable manually for the mouse samples, as they contained large global features in the form of both the CC and cingulum and even a little beyond. Despite our efforts, we could not obtain similar matching for

the monkey samples. Due to the imaging at different beamlines, the samples had been physically moved and repositioned in a new setup, thus losing alignment. This process gradually becomes easier and more streamlined as the various synchrotron beamlines develop. *Walsh et al., 2021* recently succeeded in applying several zoom-ins on a low-resolution overview scan of an intact brain within the same scanning session at the ESRF BM05 beamline. This approach removes the need to collect small selective biopsy samples for the different experiments. However, the finest pixel size in Walsh et al. was isotropic 2.5 µm, which might be too coarse to perform the structure tensor analysis that we undertook to map fibre organisation across anatomical length scales to be compared with diffusion MRI.

### Resolution

The different imaging setups each have characteristic image resolutions and native voxel sizes, but these are not the only relevant factors in this study. The targeted anatomical features are a primary consideration. Our examination of the monkey and mouse samples at the same beamlines gave approximately the same image resolution, voxel size, and FOV size. Nevertheless, the anatomical scales are quite different simply because the mouse brain is smaller, with a brain volume ratio of approximately 1:190. Thus, the same synchrotron FOV covered only a small part of the CC in the monkey, vs. a large portion of the major callosal pathway and part of the cingulum in the mouse. When defining terminologies such as 'high image resolution,' it is essential to relate it to the size of the anatomical structures of interest (*Dyrby et al., 2014*).

A limiting consequence of having samples imaged at differing anatomical scales is that certain measures become inherently hard to compare in a normalised way. The tractography-based metrics—tortuosity and maximum deviation—serve as good examples of this resolution and FOV dependence. In the ESRF samples, the anatomical scale was at the level of individual axons, and the streamline metrics primarily reflect micro-scale effects from the extra-axonal environment, such as the influence of cells and blood vessels. In comparison, the larger anatomical scale in the DESY samples represents the level of fasciculi and above, with metrics influenced by macroscopic effects, such as the bending of the CC pathway. Both scales are interesting and can provide valuable insights, but caution is required when comparing the numbers, especially for cross-species studies where there is a significant difference in brain volume ratios.

Within the same species, assuming perfect co-registration of samples, it would be possible to perform correlative imaging and analysis. This would allow validation of whether tractography streamlines could be reproduced at different image resolutions within the same normalised FOV. Although this was not possible with the current data and experimental setup, it would be an interesting point to pursue in future work.

Additionally, the image resolution does not determine voxel size, since acquired data of fixed resolution can be up- or down-sampled by interpolation. Indeed, interpolation does not change the image information (*Dyrby et al., 2014*), while the image resolution is fundamentally defined by the 'quality' of the imaging setup. As summarised in *Table 2*, we interpolated by down sampling for ease of processing and visualisation. Nonetheless, we continue referring to the datasets according to the original reconstructed voxel size (e.g. 75 nm of the ESRF data), as this reflects the approximate inherent image resolution. We minimised the effects of interpolation on the results by specifying the parameters and converting quantities to physical distances whenever possible. Furthermore, we did not target anatomical features such as axon geometries, which cannot be robustly disentangled/quantified after downsampling, although other approaches may serve that purpose (*Andersson et al., 2020*). Therefore, if the results are to be reproduced by others, then comparable imaging set-ups should be used.

## Materials and methods
### Monkey

The tissue came from a 32-mo-old female perfusion-fixated vervet (*Chlorocebus aethiops*) monkey brain, obtained from the Montreal Monkey Brain Bank. The monkey, cared for on the island of St. Kitts, had been treated in line with a protocol approved by The Caribbean Primate Center of St. Kitts.

## Mice

C57BL/6 female mice were obtained from Taconic Ltd. (Ry, Denmark). Mice were bred at the Biomedical Laboratory, University of Southern Denmark according to protocols and guidelines approved by the Danish Animal Health Care Committee (2014-15-00369). All animal experiments complied with the EU Directive 2010/63/EU for animal experiments.

Repeated oral administration of the copper chelator bis-cyclohexanone-oxalyldihydrazone (cuprizone) leads to demyelination and oligodendrocyte loss notably in the CC, and thus serves as a model for the demyelination lesions in patients with multiple sclerosis (MS) (*Torkildsen et al., 2008*). Cuprizone (Sigma Aldrich, MO, USA) was administered as 0.4% cuprizone in powdered standard chow to female mice aged 8–9 wk for 5 wk. Control mice were kept on a normal diet. During experiments, mice were weighed every second day to monitor the characteristic weight loss due to cuprizone exposure, with euthanasia of mice losing more than 20% of their baseline body weight. After 5 wk on the cuprizone diet, the mice were euthanized with an overdose of pentobarbital (Glostrup Apotek, Glostrup, Denmark) followed by perfusion with DPB and 4% paraformaldehyde (PFA). The brains were stored in 4% PFA at 4 °C.

## Diffusion MRI

Two acquisition setups were used to collect the ex vivo diffusion MRI data sets on whole brains from the monkey (N=1) and normal mouse (N=1). The protocol for the monkey brain was from *Andersson et al., 2020* and includes three shells: b-values of [2011, 2957, 9259] s/mm^2, (gradient strength (G) = [300, 219, 300] mT/m, gradient duration ($\delta$) = [5.6, 7.0, 10.5] ms, gradient separation ($\Delta$) = [12.1, 20.4, 16.9] ms), [84, 87, 68] non-collinear diffusion encoding directions and the number of b=0 s/mm$^2$ was [15, 16, 13]. The higher b-value was adjusted to the lowered ex vivo diffusivity compared to in vivo (*Dyrby et al., 2011*). Correspondingly, the mouse brain protocol is from *Perens et al., 2023* and includes a single shell: b-value of 4000 s/mm^2 (G=456 mT/m, $\delta$=5 ms, $\Delta$=13 ms), 60 non-collinear diffusion encoding directions and five b=0 s/mm$^2$. Indeed, whole-brain MRI scanning was performed on both the normal and cuprizone mice prior to synchrotron sample preparation. However, the MRI image quality was poor and could not be improved, as this was only realised after the tissue had been processed for synchrotron imaging. Therefore, the collected MRI mice data from *Perens et al., 2023* was included in the study.

Tissue preparation in both species followed a standard pipeline for diffusion MRI ex vivo (*Dyrby et al., 2011*). To reduce susceptibility artefacts and avoid air bubbles, we scanned the monkey brain in a double-sealed plastic bag filled with PBS, whereas the mouse brain was kept in the skull and placed in a sealed plastic bag with PBS. The monkey brain data was collected on an experimental 4.7 Tesla Agilent MRI scanner, whereas the mouse brain data were collected on an experimental 7 Tesla Bruker Biospec MRI scanner. We used a quadrature radio frequency volume coil for the monkey, and a 2-parallel cryo-coil probe for the mouse brains. The isotropic image voxels were 0.55 mm for the monkey brain and 0.125 mm for the mouse. The monkey brain acquisition used an optimised three-shell ActiveAx MRI protocol based on a maximal gradient strength of 300 mT/m for ex vivo tissue as in *Dyrby et al., 2013*. The mouse acquisition used a single-shell diffusion MRI protocol for diffusion tensor imaging (*Perens et al., 2023*). All whole-brain diffusion MRI data sets are available at https://www.drcmr.map/.

Before local fibre modelling, the diffusion MRI datasets were denoised (*Veraart et al., 2016*) and processed in the MRTrix3 software toolbox (RRID:SCR_006971)(*Andersson et al., 2020*; *Perens et al., 2023*) to remove Gibbs ringing artefacts (*Kellner et al., 2016*). Then, we fitted the single-fibre diffusion tensor and the multi-fibre constrained-spherical deconvolution models in the MRtrix3 software toolbox (*Tournier et al., 2019*). For the monkey data, we fitted the constrained-spherical deconvolution using only a single-shell b-value i.e., 9686 s/mm$^2$, from which we estimated the fibre orientation distribution (*Tournier et al., 2004*). The diffusion tensors were fitted with a single-shell b-value, i.e., 2957 s/mm$^2$ for the monkey (*Andersson et al., 2020*) and 4000 s/mm$^2$ for the mouse (*Perens et al., 2023*). From the diffusion tensor, we estimated the three Eigenvectors representing fibre orientations as well as the fractional anisotropy (FA) metric (*Basser et al., 1994*). Since the FA value of microstructure anisotropy is biased by fibre architecture (*Andersson et al., 2020*), we also fitted a micro-tensor model to estimate µFA from the monkey three-shell diffusion MRI data set of monkey (*Kaden et al., 2016*). The diffusion tensor model assumes a single tensor per voxel, whereas

the micro-tensor model assumes a micro-tensor regime with many micro-tensors on the diffusion length scale. Hence, the micro-tensor model is not sensitive to fibre organisation (*Kaden et al., 2016*; *Lasič et al., 2014*; *Jespersen et al., 2013*).

## Diffusion MRI human

We used data from four subjects of the Human Connectome Project (HCP) Adult Diffusion database to compute the eigenvalues and eigenvectors of the diffusion tensor (*Basser et al., 1994*) via a two-step weighted and iterated least-squares method (*Veraart et al., 2013*) as implemented in MRtrix3 (*Tournier et al., 2019*). Data were denoised as indicated in *Pizzolato et al., 2023* using a Rician variance stabilisation transform (*Foi, 2011*) in combination with PCA optimal shrinkage (*Gavish and Donoho, 2017*), with subsequent application of Gibbs ringing removal (*Kellner et al., 2016*) and eddy current distortion correction (*Andersson and Sotiropoulos, 2016*). For the estimation of the tensor, we selected only the b=0 and the 64 volume directions corresponding to b=1000 s/mm$^2$. The resulting eigenvector colour-coded maps (*Pajevic and Pierpaoli, 2000*) are shown in *Appendix 1—figure 1*, which is organised similarly to *Figure 2A*.

## Tissue preparation for synchrotron imaging

To perform the SRI experiments, small tissue samples extracted from the monkey and normal (N=1) and cuprizone (N=1) mouse brains were processed and embedded in EPON.

The monkey brain was sliced in the sagittal plane with a monkey brain matrix. Cylindrical samples of 1 mm diameter were extracted from the mid sagittal CC and the centrum semiovale (CS) with a biopsy punch. After post-fixation in 2.5% glutaraldehyde for 24 hr, they were stained by immersion in 0.5% osmium tetroxide (OsO$_4$), followed by an embedding in EPON resin and shaped into blocks measuring 1×1×4 mm. For details of sample preparation, see *Andersson et al., 2020*.

The brains of the normal and cuprizone mice were cut into 1 mm thick coronal slices using a mouse brain matrix. For the cuprizone mouse, we selected a slice where a demyelination lesion in the white matter was visible. The same slice position was selected in the normal mouse brain for comparison. After selecting the slice containing the splenium of the CC, we carefully excised the part of the splenium traversing the mid-sagittal plane and extending approximately 2 mm into the left hemisphere, using a scalpel under a microscope. The mouse brain samples were then processed as described above. Once the EPON resin had polymerized, we used a metallographic grinder to polish the blocks to have a smooth surface, a thickness of approximately 700 μm, and a length of a few millimetres.

## Synchrotron imaging

The specimens were imaged at beamline ID16A of the European Synchrotron Radiation Facility (ESRF) with x-ray nano-holotomography, as described in *Andersson et al., 2020*. In short, the samples were illuminated with a nano-focused cone (*Cesar da Silva et al., 2017*) 17 keV x-ray beam. The samples were rotated over 180 degrees, and tomographic scans were acquired at four different propagation distances (*Hubert et al., 2018*). Each scan consisted of 1800 projections with exposure times of 0.22 s, and a pixel size of 100 nm or 75 nm, taking approximately 4 hr to acquire. Upon performing the phase retrieval and tomographic reconstruction, the resulting volumes had dimensions 2048×2048 ×2048 voxels. In the case of the healthy mouse sample, the reconstruction was performed in an extended FOV providing a volume of 3200$^3$ voxels as presented in *Table 1*. For the monkey CC and CS samples, we collected four consecutive image volumes with a small overlap to extend the total FOV.

The specimens were also imaged at Deutsches Elektronen-Synchrotron (DESY), at the synchrotron radiation nano-CT end station (GINIX) of the P10/PETRA III beamline (*Salditt et al., 2015*). Here, a 13.8 keV x-ray beam illuminated the EPON samples, which were rotated through 180 degrees to acquire tomographic scans at the lens-coupled detector (XSight Micron, Rigaku). Each tomographic scan consisted of 1000 projections with 20 ms of exposure. The detector was placed in the direct-contrast regime of the sample, and a phase retrieval was performed with an in-house Bronnikov-aided correction-based algorithm (*Lohse et al., 2020*; *De Witte et al., 2009*), prior to tomographic reconstruction, which produced volumes of voxel size 550 nm and variable dimensions, as presented in *Table 1*. Only one volume was collected per sample.

Note that several separate beamline experiments were conducted to collect the volumes listed in *Table 1*. In the first two experiments, samples from the monkey brain were scanned at ESRF and

DESY, respectively. The samples from the mouse brain were imaged in two subsequent experiments. Consequently, the location of the identified demyelinating lesion in the cuprizone mice, which cannot be precisely controlled, did not match the location of the CC biopsies in the monkey.

Due to the size of the data selected, processed volumes, masks and results are available at https://zenodo.org/records/10458911. Other datasets can be shared on request.

## Image registration

Finding spatial correspondences within and across image modalities was largely a manual process. For aligning DESY/ESRF images to MRI, photo documentation of tissue extraction sites was visually compared with the dMRI scan, and the best matching dMRI voxel or region was selected manually. For registering ESRF images to DESY scans, manual initialisation in ITK-SNAP was followed by rigid registration refinement, employing rigid transformation and mutual information-based image similarity. Stitching individual ESRF fields of view involved translation-only registration, where the approximate translation shift - known from the scan setup- was manually refined using the ITK-SNAP (RRID:SCR_002010).

## Structure tensor analysis

For each synchrotron volume, we computed the 3D structure tensor for each voxel (*Jeppesen et al., 2021*; *Khan et al., 2015*). The processing involves three steps: (1) Computation of the image gradients, by filtering with the derivative of a Gaussian kernel, where the parameter σ is the standard deviation of the Gaussian; (2) Calculating the outer product of the gradient with itself, yielding a tensor in each image voxel; (3) Aggregation of tensor information within a local neighbourhood, by filtering with a Gaussian kernel i.e., also known as the patch size, where the parameter $\rho$ is the standard deviation of the Gaussian.

Similar to the Diffusion Tensor in MRI (*Basser et al., 1994*), the eigendecomposition of the structure tensor defines a 3D ellipsoid, whose axes are scaled according to the eigenvalues $\lambda_1^*$, $\lambda_2^*$ and $\lambda_3^*$ (where $\lambda_1^* > \lambda_2^* > \lambda_3^*$ and may be normalised to ensure that $\sum \lambda_i^* = 1$), and their orientations are defined by the three orthogonal eigenvectors $v_1$, $v_2$, and $v_3$. Throughout this paper, the estimated structure tensor decomposition is by default converted to a diffusion-like tensor, as the structure tensor and the diffusion tensor are 'inverted' to one another (*Khan et al., 2015*). We use the following single parameter ($\gamma$) model for converting the eigenvalues of the structure tensor:

$$\lambda_i = e^{-\lambda_i^*/\gamma}$$

Afterwards, the values are normalised to have $\sum \lambda_i = 1$.

From the eigenvalues, the anisotropy of the structure tensor i.e., its shape is characterised by the same Fractional Anisotropy (FA) metrics as defined for the diffusion tensor (*Basser et al., 1994*) i.e.,

$$FA = \sqrt{\frac{3}{2}\left(\frac{\left(\lambda_1 - \hat{\lambda}\right)^2 + \left(\lambda_2 - \hat{\lambda}\right)^2 + \left(\lambda_3 - \hat{\lambda}\right)^2}{\lambda_1^2 + \lambda_2^2 + \lambda_3^2}\right)}$$

where $\hat{\lambda} = \left(\lambda_1 + \lambda_2 + \lambda_3\right)/3$ is the mean of the three eigenvalues.

We introduce a scale-space structure tensor approach to ease parameter tuning and provide a scale-invariant analysis. The variation of axon diameters within some of the samples makes it challenging to capture all relevant structural orientation information with a single set of (σ, $\rho$)-parameters, i.e., for a single patch size. Therefore, we employed a 'scale space structure tensor' approach (*Lindeberg, 1998*). Here, the structure tensor is computed multiple times using a suite of varying (σ, $\rho$)-parameters called *scales*. Finally, for each voxel, we search for the dominating scale from which we retain the tensor. The criterion for selecting the dominant scale is based on the scale-wise relative maximum fractional anisotropy (FA) value. Let $FA_j^i$ be the calculated FA-value in voxel $j$ at scale $i$ using the parameter set (σ, $\rho$)$^i$. The dominant scale index, $d_j$, for voxel $j$ is then selected as

$$d_j = \underset{i}{\operatorname{argmax}} \left( \frac{FA_j^i}{\underset{j}{\max}\left(FA_j^i\right)} \right) \; , \; j \; = \; 1, 2, ..., N$$

In other words, at a given scale, the maximum observed FA value across all voxels, $\underset{j}{\max}\left(FA_j^i\right)$, is used as a normalisation factor. With this approach, we select the structure tensor that provides the most anisotropic response across the set of manually defined scales. This ensures that the tensor-derived quantities—eigenvectors, eigenvalues, and FA—become independent of the axon diameter. Using the FA value for scale selection is reasonable, as we focus on the fibre-like axons. *Appendix 1— figure 2* in the Appendix illustrates the scale-space structure tensor concept applied to the monkey CC ESRF data, where the FA of three selected scales of gradually decreasing patch sizes (scales 2, 6, and 8), are shown to emphasise distinctly different microstructural features. The dominant scale illustration shows how scales with large patch sizes are selected where the axons are large.Structure tensor-based quantifications.

The ST analysis, whether using the scale space or standard variant with a single scale, provides one tensor per voxel. Each sample is then quantified by the statistical distribution of FA values and the FOD based on the principal directions. The distributions are estimated from the combination of the following inclusion and exclusion regions of interest (ROIs).

- Inclusion ROIs: In the mouse samples, ROIs defining the CC and the cingulum were generated, allowing for FA and FOD quantifications for each pathway separately.
- Exclusion ROIs: In all samples, an ROI representing non-stained or non-tissue regions within the FOV was used to exclude those voxels. In the monkey samples, an ROI representing blood vessels was further applied for additional voxel exclusion. Notably, including blood vessels in the results had minimal influence, as they constituted a small percentage of the volume and it would not alter any conclusions.

The ROI segmentation masks were generated either manually using ITK-SNAP RRID:SCR_002010 or through intensity thresholding followed by morphological operations.

**The statistical FA distributions** were generated based on the selected voxels by making a kernel density estimation of the probability density function over the FA values.

**The Fibre Orientation Distribution** (FOD) was generated by binning the principal (unit) vectors of all selected voxels into a spherical histogram parametrized by azimuth and elevation angles. The poles for this representation are poorly defined and hard to visualise. Therefore, the polar direction is aligned along the anatomical axis where we expect the least directional contribution for the particular sample. Additionally, since the bins in the parameterisation do not have equal areas, the actual area is estimated and used to normalise the histogram, obtaining a fibre orientation based probability density function.

To characterise the FOD, we fitted a spherical Bingham distribution to the FOD using the mtex toolbox (https://mtex-toolbox.github.io/BinghamODFs.html). From the Bingham parameters, we derive two indices: the Orientation Dispersion Index (ODI), which describes the dispersion of the fibres on the surface of the unit sphere, and the Dispersion Anisotropy Index (DA), which expresses the anisotropic shape of the fibre dispersion (*Tariq et al., 2016*). Both metrics range from 0 to 1. A low ODI indicates a population of fibres with a narrow spread (a focused main direction), while a low DA suggests that the shape of the fibre dispersion is close to isotropic. It should be noted that the use of ODI and DA is only meaningful for a single pathway FOD.

## Structure tensor-based tractography

The principle and the process of structure tensor-based tractography are much the same as in diffusion MRI-based tractography, but applied to the synchrotron volumes using the direction vectors of the structure tensor and the deterministic *FACT* algorithm (*Tournier et al., 2019*). The process is controlled by defining **seeding point regions**, **masks** for rejection, inclusion/termination, and various **streamlined filtering** parameters, such as minimum length. We utilise an a priori anatomical understanding of the sample and it's the axonal organisation to manually generate the seeding region(s). For example, in the CC samples, the axons should primarily run in the L-R direction. We therefore create a mask at the left and right ends of the sample (with some margin from the sample edge).

When seeding points in the left region of interest (ROI), streamlines must reach the right ROI to be included, and vice versa.

In several instances, we employed inclusion or rejection masks, which defined the permitted boundaries of travel for the streamlines. In essence, these masks should roughly represent axonal tissue segmentations. Depending on the sample, such a mask can be generated by thresholding either FA - or image intensity values, followed by morphological operations (opening, closing) to close holes and remove small spurious regions.

## Streamline clustering

The output of tractography is unstructured, thus often resulting in an overabundance of streamlines, which may be hard to interpret. It is then beneficial to apply a streamline clustering method, which can collect multiple streamlines into meaningful axonal bundles/fasciculi, while filtering away lone and spurious streamlines. To this end, we used the QuickBundles method (*Garyfallidis et al., 2012*), known for its simplicity and scalability, with manual selection of the distance threshold parameter for each sample individually.

## Streamline analysis

We use two different metrics to quantify the clustered tractography streamlines: Tortuosity and maximum deviation.

The tortuosity index is a single number representing the non-straightness of the trajectory of each streamline. It is calculated as the ratio of the length of a straight line, $d$, (between the streamline endpoints) and the piecewise length of the actual streamline trajectory, $L$.

$$\tau = L/d$$

The tortuosity index is bound between $[1, \infty]$. Following this definition, perfectly straight streamlines have a score of 1, and erratic streamlines have higher values. The various tortuosity indices are finally summarised in a histogram for each sample.

The maximum deviation is a supplementary index that describes for each streamline the largest observed physical deviation from a straight line between the given endpoints, which might be described as the maximum amplitude. This index may be easier to interpret in anatomical terms, compared to the unit-less tortuosity.

Similar to the tortuosity measure, we define the direct vector, $v_d$, as going from the streamline starting point to the end point, i.e., $\|v_d\| = d$. Additionally, we let $v_i$ be the vector extending from the streamline starting point to either of the $N$ streamline sampling points. We then measure the orthogonal distance from each sample point of the streamline to this direct vector by finding the projection of $v_i$ onto $v_d$. The maximum deviation, $d_{max}$, is then defined as the largest observed distance between all sample points and $v_d$,

$$d_{max} = \underset{i}{\operatorname{argmax}} \left( \left\| v_i - \frac{v_i \cdot v_d}{v_d \cdot v_d} v_d \right\| \right), \quad i = 2 : N - 1$$

Statistical tests of significance were applied to both described streamline measures where relevant. Specifically, we conducted:

- Two-sample Kolmogorov-Smirnov tests to assess whether two selected distributions are identical. The null hypothesis is that the two selected distributions are equal (*Massey, 1951*).
- Two-sided Wilcoxon rank sum tests to compare the medians of two selected distributions. The null hypothesis is that the median values are equal (*Hollander et al., 2015*).
- Brown-Forsythe tests to examine the variance of the two selected distributions. The null hypothesis is that the distributions have equal variance (*Brown and Forsythe, 1974*).

All tests were performed at a significance level of 0.05, and we used the standard implementations in MATLAB 2021b (MathWorks, Massachusetts, USA).

# Acknowledgements

We thank Susanne Sørensen for her assistance with the tissue preparation, and Johanna Perens from Gubra A/S for preparing the mouse MRI data. The authors acknowledge Professor Paul Cumming for critical reading of the manuscript. We acknowledge DESY (Hamburg, Germany), a member of the Helmholtz Association HGF, for the provision of experimental facilities. Parts of this research were carried out at PETRA III and we would like to thank Dr. Michael Sprung for assistance in using the GINIX setup at P10. Beamtime was allocated for proposal(s) I-20170269 EC and I-20180267 EC. We acknowledge the European Synchrotron Radiation Facility (ESRF) for the provision of synchrotron radiation facilities under proposal numbers LS-2702 and LS-2840 and we would like to thank Peter Cloetens for assistance and support in using beamline ID16A. MA and HMK were supported by Capital Region Research Foundation Grant A5657 (principal investigator: TBD). MLE is grateful for the financial support from Lundbeckfonden (R347-2020-2454). ZI is grateful for the financial support from Lundbeckfonden R118-A11472, Scleroseforeningen A41354, and Independent Research Fund Denmark (DFF 9039-00370B). The project has received funding from the European Research Council (ERC) under the European Union's Horizon Europe research and innovation programme (grant agreement No. 101044180) (Principal Investigator: TBD).

The human MRI Data were in part provided by the Human Connectome Project, WU-Minn Consortium (Principal Investigators: David Van Essen and Kamil Ugurbil; 1U54MH091657) funded by the 16 NIH Institutes and Centers that support the NIH Blueprint for Neuroscience Research; and by the McDonnell Center for Systems Neuroscience at Washington University.

## Additional information

### Funding

| Funder | Grant reference number | Author |
|---|---|---|
| Captital Region of Denmark Research Foundation | A5657 | Hans Martin Kjer Mariam Andersson |
| European Research Council | 101044180 | Tim B Dyrby |
| Lundbeck Foundation | R118-A11472 | Zsolt Illes |
| Independent Research Fund Denmark | 9039-00370B | Zsolt Illes |
| Scleroseforeningen | A41354 | Zsolt Illes |
| Lundbeck Foundation | R347-2020-2454 | Maria Louise Elkjær |

The funders had no role in study design, data collection and interpretation, or the decision to submit the work for publication.

### Author contributions

Hans Martin Kjer, Data curation, Software, Formal analysis, Validation, Investigation, Visualization, Methodology, Writing – original draft, Writing – review and editing; Mariam Andersson, Data curation, Validation, Investigation, Visualization, Methodology, Writing – original draft, Writing – review and editing; Yi He, Alessandro Daducci, Marco Pizzolato, Anna-Lena Robisch, Marina Eckermann, Mareike Töpperwien, Anders Bjorholm Dahl, Maria Louise Elkjær, Maurice Ptito, Vedrana Andersen Dahl, Methodology, Writing – review and editing; Alexandra Pacureanu, Resources, Data curation, Methodology, Writing – review and editing; Tim Salditt, Zsolt Illes, Resources, Methodology, Writing – review and editing; Tim B Dyrby, Conceptualization, Resources, Supervision, Funding acquisition, Validation, Investigation, Methodology, Writing – original draft, Project administration, Writing – review and editing

### Author ORCIDs

Hans Martin Kjer ⓘ https://orcid.org/0000-0001-7900-5733
Tim Salditt ⓘ https://orcid.org/0000-0003-4636-0813

Tim B Dyrby ⓘ https://orcid.org/0000-0003-3361-9734

### Ethics

The monkey, cared for on the island of St. Kitts, had been treated in line with a protocol approved by The Caribbean Primate Center of St. Kitts.Mice were bred at the Biomedical Laboratory, University of Southern Denmark according to protocols and guidelines approved by the Danish Animal Health Care Committee (2014-15-00369). All animal experiments complied with the EU Directive 2010/63/EU for animal experiments.

Reviewer #1 (Public Review): https://doi.org/10.7554/eLife.94917.3.sa1
Reviewer #2 (Public Review): https://doi.org/10.7554/eLife.94917.3.sa2
Author response https://doi.org/10.7554/eLife.94917.3.sa3

### Data availability

The processed tomographic volumes used to derive the results of the paper are available at the following Zenodo repository: https://zenodo.org/records/10458911. These volumes are in an accessible format, compatible with most standard medical imaging viewing software. Additionally, several intermediate results and masks are provided to enable easy interaction and access to the 3D experience of the data, which can be challenging to convey in 2D printed figures. For additional descriptive details about the data files, links to code, methodology, and software viewers, please refer to this link.Access to versions of the tomographic image data at earlier processing stages, such as the native reconstructed tomograms or the raw projection images, can be accommodated upon request by contacting the corresponding authors. This raw data has not yet been shared in open data repositories due to its massive size and the specialized, beamline-dependent hardware and software required to interact with such datasets. We are happy to share and provide guidance for researchers interested in these types of data.

The following dataset was generated:

| Author(s) | Year | Dataset title | Dataset URL | Database and Identifier |
|---|---|---|---|---|
| Kjer HM, Andersson M, He Y, Pacureanu A, Daducci A, Pizzolato M, Salditt T, Robisch AL, Eckermann M, Toepperwien M, Dahl AB, Elkjaer ML, Illes Z, Ptito M, Dahl VA, Dyrby TB | 2024 | White matter multi-scale dataset: Diffusion weighted MRI and synchrotron x-ray scans of vervet monkey-, healthy mouse-, and cuprizone mouse brains | https://doi.org/10.5281/zenodo.10458911 | Zenodo, 10.5281/zenodo.10458911 |

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

## Appendix 1

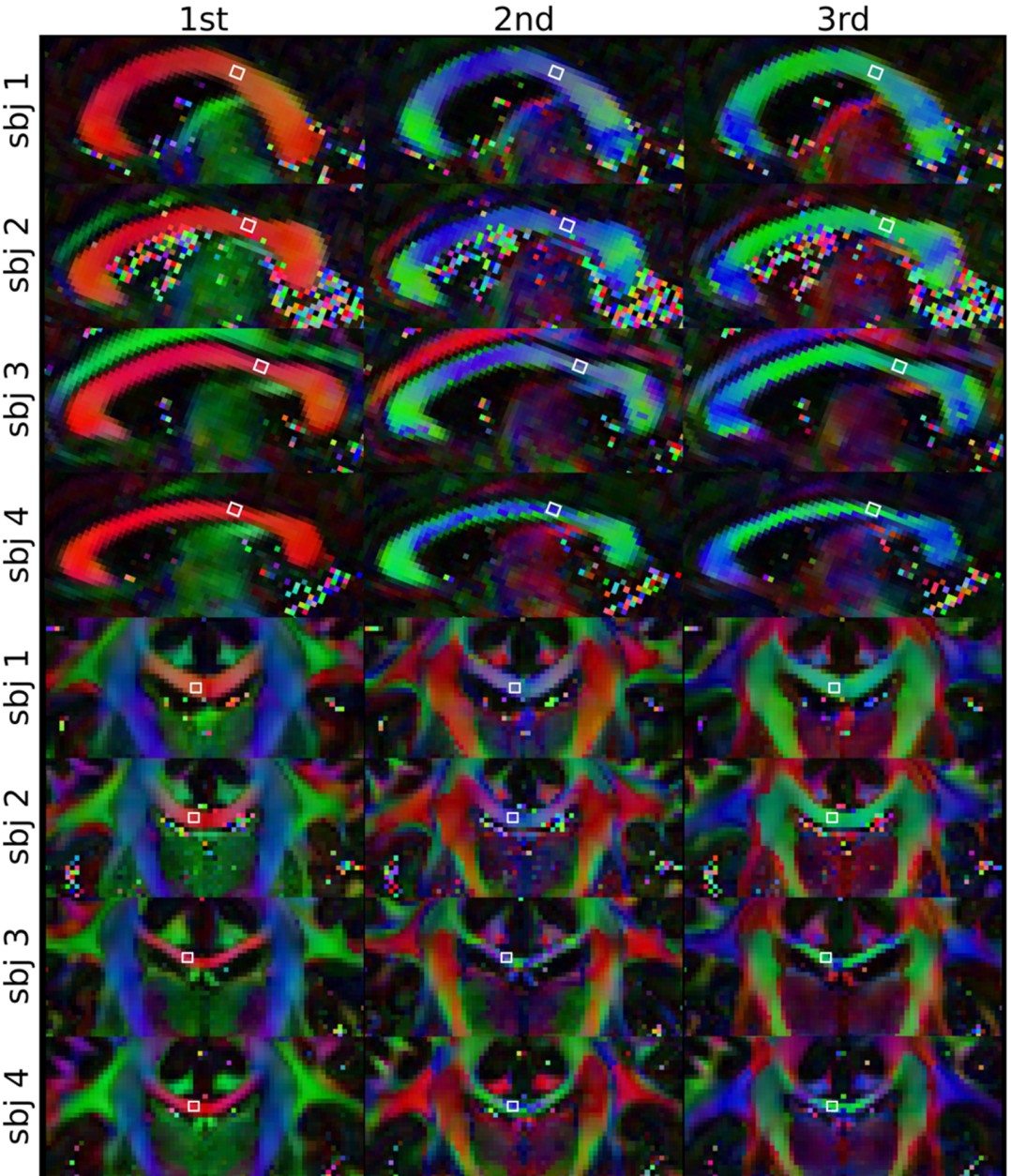

**Appendix 1—figure 1.** Human Connectome Project (HCP) dataset diffusion tensor modelling. The RGB colour-coded first, second, and third eigenvectors (ordered by decreasing eigenvalues) for the four subjects in both sagittal and coronal crop-out views.

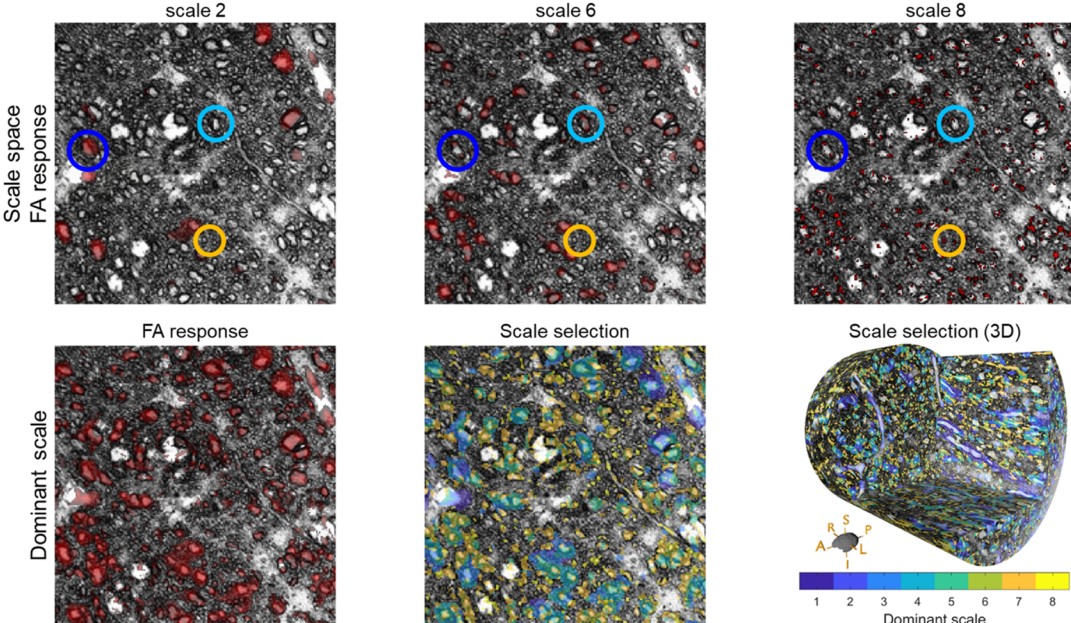

**Appendix 1—figure 2.** Example of applying the scale-space structure tensor. (Top row): A region of a slice from the European Synchrotron (ESRF) monkey corpus callosum (CC) sample, where the fractional anisotropy (FA) is estimated from eight scales (see *Table 2*), here showing only the scales 2, 6, and 8. The red transparent overlay shows the thresholded FA response (FA > 0.7). Notice how large, medium, and small cross-sectional axons respond differently at the different scales (represented by the blue, light blue, and orange circles, respectively). (Bottom row, left): The final thresholded dominant FA response (dFA > 0.7), shows high FA values for all axons regardless of the diameter. (Bottom row, mid and right): The same subset of voxels coloured according to the dominant scale index for the same 2D slice region and in a 3D rendering respectively. Notice how the low scales (large kernels) are selected for large axons, and similarly high scales (small kernels) are selected for the small axons.

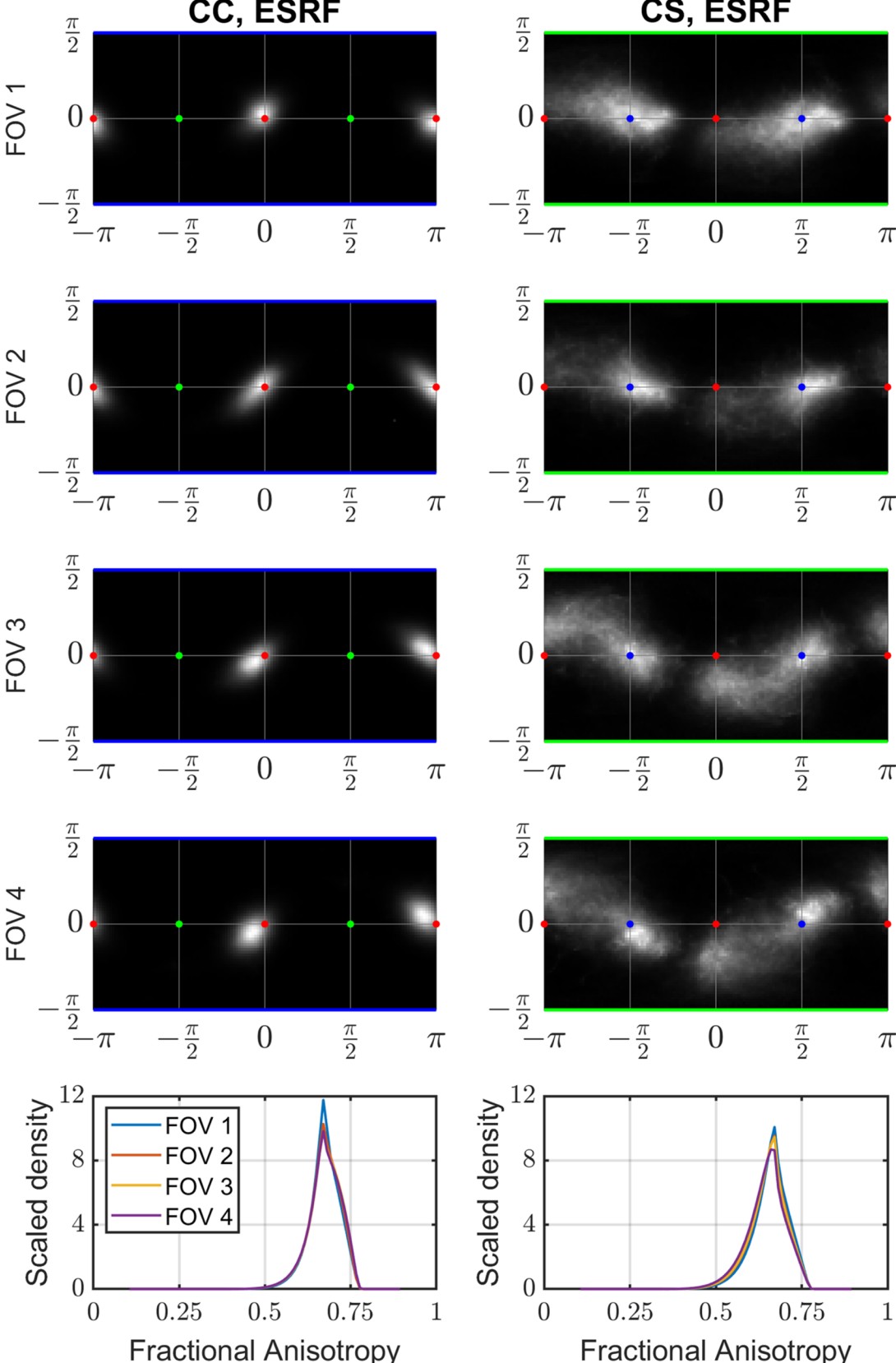

**Appendix 1—figure 3.** ST-derived statistics from the individual stacked field-of-views (FOVs) in the monkey European Synchrotron (ESRF) corpus callosum (CC) and *centrum semiovale* (CS) sample. (Rows 1-4): Individual *Appendix 1—figure 3 continued on next page*

*Appendix 1—figure 3 continued*
fibre orientation distributions (FODs) showing minor variation as the FOV placement changes. Across the CC, the most notable change is the anisotropic shape of the peak. Across the CS, the prominence of the L-R directed peak (near the red points) represents the largest variation. (Row 5): FA distributions of the four individual FOVs plotted together, showing almost no difference in the statistics.

