## [Editor Report · eLife Assessment]

This **valuable** study presents new observations on white matter organisation at the micron scale, using a combination of synchrotron imaging and diffusion MRI across two species. Notably, the authors provide **solid** evidence for the fasciculation of axons within major fibre bundles into laminar structures, though these structures are not consistently observed across modalities or species. The study will be of general interest to neuroanatomists and those interested in white matter imaging.

---

## [Referee Report · Reviewer #1 (Public Review)]

This study presents valuable observations of white matter organisation from diffusion MRI and two types of synchrotron imaging in both monkeys and mice. Cross-modality comparisons are interesting as the different methods are able to probe anatomical structures at different length scales, from single axons in high-resolution synchrotron (ESRF) imaging, to clusters of axons in lower-resolution synchrotron (DEXY) data, to axon populations at the mm-scale in diffusion MRI. By acquiring all modalities in monkey and mouse ex vivo samples, the authors can observe principles of fibre organisation, and characterise how fibre characteristics, such as tortuosity and micro-dispersion, vary across select brain regions and in healthy tissue versus a demyelination model.

One very interesting result is the observation of apparent laminar organisation of fibres in ex vivo monkey white matter samples. DESY data from the corpus callosum shows fibres with two dominant orientations (one L-R, one slightly inclined), clustered in laminar structures within this major fibre bundle. Thanks to the authors providing open data, I was able to look through the raw DESY volume and observe regions with different "textures" (different orientations) in the described laminar arrangement. That this organisation can be observed by eye, as well as by structure tensor, is fairly convincing.

---

## [Referee Report · Reviewer #2 (Public Review)]

Summary:

In this work, the authors combine diffusion MRI and high-resolution x-ray synchrotron phase-contrast imaging in monkey and mouse brains to investigate the 3D organization of brain white matter across different scales and species. The work is at the forefront of the anatomical investigation of the human connectome and aligns with several current efforts to bridge the resolution gap between what we can see in vivo at the millimeter scale and the complexity of the human brain at the sub-micron scale. The authors compare the 3D white matter organization across modalities within 2 small regions in one monkey brain (body of the corpus callosum, centrum semiovale) and within one region (splenium of the corpus callosum) in healthy mice and in one murine model of focal demyelination. The study compares measures of tissue anisotropy and fiber orientations across modalities, performs a qualitative comparison of fasciculi trajectories across brain regions and tissue conditions using streamlined tractography based on the structure tensor, and attempts to quantify the shape of fasciculi trajectories by measuring the tortuosity index and the maximum deviation for each reconstructed streamline. Results show measures of anisotropy and fiber orientations largely agree across modalities, especially for larger FOV data. The high-resolution data allows us to explore the fiber trajectories in relation to tissue complexity and pathology. The authors claim the study reveals new common organization principles of white matter fibers across species and scales, for which axonal fasciculi arrange into sheet-like laminar structures.

Strengths:

The aim of the study is of central importance within present efforts to bridge the gap between macroscopic structures observable in vivo in humans using conventional diffusion MRI and the microscopic organization of white matter tissue. Results obtained from this type of study are important to interpret data obtained in vivo, inform the development of novel methodologies, and expand our knowledge of the structural and thus functional organization of brain circuits.

Multi-scale data acquired across modalities within the same sample constitute extremely valuable data that is often hard to acquire and represent a precious resource for validation of both diffusion MRI tractography and microstructure methods.

The inclusion of multi-species data adds value to the study, allowing the exploration of common organization principles across species.

The addition of data from a murine cuprizone model of focal demyelination adds interesting opportunities to study the underlying biological changes that follow demyelination and how these impact tissue anisotropy and fiber trajectories. These data can inform the interpretation and development of diffusion MRI microstructure models.

[Editors' note: The Reviewing Editor considers that the authors addressed the reviewers' questions adequately. The original reviews are here: https://elifesciences.org/reviewed-preprints/94917/reviews]

---

## [Author Response]

The following is the authors’ response to the original reviews.

**Reviewer #1 (Public Review):**
This study presents valuable observations of white matter organisation from diffusion MRI and two types of synchrotron imaging in both monkeys and mice. Cross-modality comparisons are interesting as the different methods are able to probe anatomical structures at different length scales, from single axons in high-resolution synchrotron (ESRF) imaging, to clusters of axons in lower-resolution synchrotron (DEXY) data, to axon populations at the mm-scale in diffusion MRI. By acquiring all modalities in monkey and mouse ex vivo samples, the authors can observe principles of fibre organisation, and characterise how fibre characteristics, such as tortuosity and micro-dispersion, vary across select brain regions and in healthy tissue versus a demyelination model. The results are solid, though some statements (in the abstract/discussion) do not appear to be fully supported, and statistical tests would help confirm whether tissue characteristics are similar/different between different conditions.

R1.1: Thank you for the kind feedback. We have included statistical tests in the paper for tissue characteristics where appropriate.

Due to the very high number of sample points (one per voxel) within the 3D synchrotron volumes, testing for statistical significance is challenging for the structure tensor-based tissue fractional anisotropy (FA) metric. This causes any standard statistical test to have sufficient power to evaluate even minute differences between the volumes as statistically significant with high confidence. In other words, the null hypothesis (H0) will always be rejected with p = 0, regardless of the practical significance of the difference. Therefore, we have not added statistical analysis for FA results.

For the tractography based metrics, the number of sample points (one per streamline) is not as high as that for the structure tensor FA, thus making it more reasonable to test for statistical significance. The statistical analyses performed included tests for equality of distributions (Two-sample Kolmogorov-Smirnov tests), equality of medians (Two-sided Wilcoxon rank sum tests), and equality of variance (Brown-Forsythe tests). The results are described in relation to Figure 5(B, D), Figure 8(CF), and detailed in the Methods section.

One very interesting result is the observation of apparent laminar organisation of fibres in ex vivo monkey white matter samples. DESY data from the corpus callosum shows fibres with two dominant orientations (one L-R, one slightly inclined), clustered in laminar structures within this major fibre bundle. Thanks to the authors providing open data, I was able to look through the raw DESY volume and observe regions with different "textures" (different orientations) in the described laminar arrangement. That this organisation can be observed by eye, as well as by structure tensor, is fairly convincing. As not all readers will download the data themselves, the manuscript could benefit from additional figures/videos to demonstrate (1) the quality of the DESY data and (2) a more 3D visualisation of the laminar structures (where the coronal plane shows convincing columnar structure or stripes). Similarly in Figure 5A, though this nicely depicts two populations with different orientations, it is somewhat difficult to see the laminar structure in the current image.ESRF data of the centrum semiovale (CS) contributes evidence for similar laminar structures in a crossing fibre region, where primarily AP fibres are shown to cluster in 3 laminar structures. As above, further visualisations of the ESRF volume in the CS (as shown in Figure 4E) would be of value (e.g. showing consistency across the 4 volumes, 2D images showing stripey/columnar patterns along different axes, etc).

R1.2: Conveying complex 3D geometry through 2D still images is indeed challenging, and we greatly appreciate the reviewer’s comments and suggestions. To better communicate the understanding of the 3D anatomical environments, we have taken the following actions:

(1) To enhance insights into the tractography results in Figures 5A and 5D, we have rendered and added animations of the tractography scenes as supplemental material.

(2) To visually support 3D insights concerning the consistency of the laminar organisation of the callosal fibres, we have replaced the 2D slice views in Figures 3A and 3B with 3D renderings similar to the one in Figure 4E.

(3) An animation of Figure 4E was created to display the colour-coded structure tensor directions of all four stacked scans. This animation visually supports the complexity of the fibre orientation and the layered structural laminar organisation of the CS sample.

A key limitation of this result is that, though the DESY data from the CC seems convincing, the same structures were not observed in high-resolution synchrotron (ESRF) data of the same tissue sample in the corpus callosum. This seems surprising and the manuscript does not provide a convincing explanation for this inconsistency. The authors argue that this is due to the limited FOV of the ESRF data (~200x200x800 microns). However, the observed laminar structures in DESY are ~40 microns thick, and ERSF data from the CST suggests laminar thicknesses in the range of 5-40 microns with a similar FOV. This suggests the ERSF FOV would be sufficient to capture at least a partial description of the laminar organisation. Further, the DESY data from the CC shows columnar variations along the LR axis, which we might expect to be observed along the long axis of the ESFR volume of the same sample. Additional analyses or explanations to reconcile these apparently conflicting observations would be of value. For example, the authors could consider down-sampling the ESRF data in an appropriate manner to make it more similar to the DESY data, and running the same analysis, to see if the observed differences are related to resolution (i.e. the thinner laminar structures cluster in ways that they look like a thicker laminar structure at lower resolution), or crop the DESY data to the size of the ESRF volume, to test whether the observed differences can be explained by differences in FOV. Laminar structures were not observed in mouse data, though it is unclear if this is due to anatomical differences or somewhat related to differences in data quality across species.

R1.3: We have clarified and expanded upon the results regarding the laminar organisation observed in the monkey CC DESY data. As noted in R1.2, we replaced the 2D images in Figures 3A (DESY) and 3B (ESRF) with 3D renderings to better display the spatial outline of the laminar organisation in the volumes. The reviewer is correct that, although the smaller field of view (FOV) of the ESRF data should allow us to at least partially capture parts of the laminar organisation observed in the larger FOV of the DESY data, this is not guaranteed. It depends on how the smaller FOV is positioned relative to the structural organisation, and since we lack co-registration, we do not know this. It should now be visually evident that the ESRF FOV can be placed such that it does not cover the observed laminae, a point which is now also emphasised in the Discussion.

Secondly, it is important to emphasise that the voxel colouring using the primary structure tensor direction is just a visualisation technique, which has limitations when it comes to assessing laminar organisation. Mapping 3D directions to RGB colours is inherently difficult and will always have ambiguities. If we had used the standard R-G-B to LR-AP-IS colouring in Figure 3, the laminar organisation would not be evident. Additionally, the laminae will only be visible when there are clear angular differences. There can still be a layered organisation even if we don’t observe it, which is the case for the mouse. The primary direction differences of these layers could be very low (i.e., parallel layers), and consequently not visually evident. This point has been clarified in both the Results and Discussion sections.

Finally, in response to R1.6, we have added analyses regarding the shape of the FOD, specifically estimating the Orientation Dispersion Index (ODI) and Dispersion Anisotropy (DA). This provides further context to the reviewer’s comments about the discrepancies in laminar organisation. We have reflected on the relationship between DA and the visually observed laminar organisation, and this has been integrated into the relevant parts of the Results and Discussion sections.

The changes to manuscript reflecting the statements above are listed here:

The Discussion section (page 21): “In the monkey CC DESY data, which has a field of view (FOV) comparable to a dMRI voxel, a columnar laminar organisation at a macroscopic level was visually revealed from the structure tensor (ST) direction colouring. However, this laminar organisation was not visible in the higher-resolution ESRF data for the same tissue sample. Although the two samples were not co-registered, the size of a single ESRF FOV within the DESY sample is illustrated in Fig. 3A. This demonstrates the possibility of placing the ESRF sample where the observed laminar structure is absent. Consequently, knowledge of the tissue structural organisation and its orientation is important to fully benefit from the stacked FOV of the ESRF sample and when choosing appropriate minimal FOV sizes in future experiments.

Interestingly, when characterising FODs with measures like ODI and DA as indicators of fibre organisation, rather than relying on visualisation, results from large- and small-FOV data show no discrepancies. This statistical approach discards the spatial context (visually perceived as laminae), highlighting the need to combine both methods.”

The Results section (page 8): “The mid-level DA values suggest some anisotropic spread of the directions, reflecting the angled laminar organisation observed in the DESY sample. Interestingly, the DA value for the ESRF sample is almost identical, despite the laminar bands being less visually apparent.”

The Results section (page 17): “Nevertheless, visualisation of orientations did not reveal any axonal organisation in the mouse CC due to the lack of local angular contrast, unlike the clear laminar structures seen in the monkey sample (Fig. 3A). Any parallel organisation in tissue remains undetectable because our visual contrast relies on angular differences.”

The Discussion section (page 22): “In the monkey CC (mid-body), we observed laminar organisation indicated by clear spatial angular differences in the ST directions in the sample (Fig. 3A). Quantifications of the FOD shape showed DA indices of 0.55 and 0.59 for the DESY and ESRF samples, respectively. In contrast, the mouse CC (splenium) did not visually reveal a similar angled laminar organisation (Fig. 7C), and the DA indices were lower, at 0.49 and 0.32, respectively. Two possible explanations exist. First, the within-pathway laminar organisation may not be identical across the entire CC. Consequently, more scans from other CC regions would be required to confirm. Second, the different species might account for the differences. Larger brains like the monkey might foster a different level of within-pathway axon organisation compared to the smaller mouse. Although we could not visually detect laminar organisation from the colour coding of the ST direction in the mouse, the non-zero DA values suggest some level of organisation. This is supported by our streamline tractography, which indicates a vertical layered organisation (Fig. 8A, B). It further aligns with studies using histological tracer mapping that shows a stacked parallel organisation of callosal projections in mice, between cortex regions M1 and S1 (Zhou et al. 2013). Nevertheless, we cannot rely solely on voxel-wise ST directions to fully describe axonal organisation, as this method does not contrast almost parallel fasciculi (inclination angles approaching 0 degrees). Analysing patterns in tractography streamlines would be an interesting future direction for this purpose.”

The authors further quantify various other characteristics of the white matter, such as micro-dispersion, tortuosity, and maximum displacement. Notably, the microscopic FA calculated via structure tensor is fairly consistent across regions, though not modalities. When fibre orientations are combined across the sample, they are shown to produce similar FODs to dMRI acquired in the same tissue, which is reassuring. As noted in the text, the estimates of tortuosity and max displacement are dependent on the FOV over which they are calculated. Calculating these metrics over the same FOV, or making them otherwise invariant to FOV, could facilitate more meaningful comparisons across samples and/or modalities.

R1.4: This raises an interesting point about the necessity of normalising the FOV to obtain invariant, tractography-based metrics of tortuosity and maximum deviation across different samples and modalities. In general, achieving this is challenging, and in this study, it is practically not possible. Between species, we encounter significant differences in brain volume ratios, which complicates the establishment of a common reference FOV due to the distinct anatomical organisation of monkey and mouse brains (see our response to R1.8). Within species, we would encounter challenges due to missing contrast—such as issues with staining—and the lack of perfect co-registration.

The Discussion section (page 28) has been extended to reflect this: ”Within the same species, assuming perfect co-registration of samples, it would be possible to perform correlative imaging and analysis. This would allow validation of whether tractography streamlines could be reproduced at different image resolutions within the same normalised FOV. Although this was not possible with the current data and experimental setup, it would be an interesting point to pursue in future work.”

Though the results seem solid, some statements, particularly in the abstract and discussion, do not seem to be fully supported by the data. For example, the abstract states "Our findings revealed common principles of fibre organisation in the two species; small axonal fasciculi and major bundles formed laminar structures with varying angles, according to the characteristics of major pathways.", though the results show "no strong indication within the mouse CC of the axonal laminar organisation observed in the monkey". Similarly, the introduction states: "By these means, we demonstrated a new organisational principle of white matter that persists across anatomical length scales and species, which governs the arrangement of axons and axonal fasciculi into sheet-like laminar structures." Further comments on the text are provided below.

R1.5: We understand that it can be misunderstood that the laminar organisation is identical in monkeys and mice, which is not the case. For example, we show that in the corpus callosum, pathways are parallel in the mouse but not in the monkey. We have clarified that while the principle of layered laminar organisation of pathways is shared between monkeys and mice, species-specific differences do exist.

We have made the following clarifying changes to the manuscript:

The Abstract (page 2): “Our findings revealed common principles of fibre organisation that apply despite the varying patterns observed across species”

The Introduction (page 4-5): “Through these methods, we demonstrated organisational principles of white matter that persists across anatomical length scales and species. These principles govern the organisation of axonal fasciculi into sheet-like laminar shapes (structures with a predominant planar arrangement). Interestingly, while these principles remain consistent, they result in varied structural organisations in different species.”

The Discussion (page 21): “despite species differences”.

One observation not notably discussed in the paper is that the spherical histograms of Figure 3E/H appear to have an anisotropic spread of the white points about 0,0. It would be interesting if the authors could comment on whether this could be interpreted as the FOD having asymmetric dispersion and if so, whether the axis of dispersion relates to the fibre orientations of the laminar structures.

R1.6: That is a good point, and to address it, we have fitted spherical Bingham distributions to the FODs, allowing us to quantify their shapes. From each Bingham distribution, we derived two wellknown indices from the diffusion MRI community: the Orientation Dispersion Index (ODI) and Dispersion Anisotropy (DA) index. The ODI explains the dispersion of fibres for a single bundle FOD, whereas DA expresses the shape of the FOD on the unit sphere surface, i.e., the degree of anisotropy. We have integrated the Bingham-based analysis into the Methods, Discussion, and Results sections concerning Figures 3 and 7, but not Figure 4, which contains multiple fibre bundles that we cannot separate on a voxel level. The analysis does not impact the overall message and conclusion but adds interesting context to the discussion around laminar organisation.

A limitation of the study is that it considers only small ex vivo tissue samples from two locations in a single postmortem monkey brain and slightly larger regions of mouse brain tissue. Consequently, further evidence from additional brain regions and subjects would be required to support more generalised statements about white matter organisation across the brain.

R1.7: Collecting more samples from various locations in the brain would provide valuable insights into the consistency of white matter organisation across anatomical length scales, as well as the structuretensor based anisotropy and tortuosity metrics. However, being awarded beamtime at two different synchrotron facilities to scan the same sample with different imaging setups is practically challenging. At the ESRF, we have gathered additional image volumes from other white matter regions of the monkey brain that support all our findings, which will be published separately. X-ray synchrotron imaging technology is advancing rapidly, with faster acquisition times enabling more image volumes to be stitched together. This extends the FOV and allows for a more robust statistical description of the anatomy. Consequently, future studies with an extended FOV and varying image resolutions could utilise a single synchrotron facility to collect additional samples, further supporting our findings.

The Discussion section (page 27) has been extended to reflect this: “Increasing the number of samples across both species and examining laminar organisation at various length scales in more regions would strengthen our findings. However, securing beamtime at two different synchrotron facilities to scan the same sample with varying image resolutions is a limiting factor. Beamline development for multiresolution experimental setups, along with faster acquisition methods, is a rapidly advancing field. For instance, the Hierarchical Phase-Contrast Tomography (HiP-CT) imaging beamline at ID-18 at the ESRF, enables multi-resolution imaging within a single session to address this challenge, though it is currently limited to a resolution of 2.5 μm (Walsh et al. 2021).”

Given the monkey results, the mouse study (section 2.5 onwards) lacks some motivation. In particular, it is unclear why a demyelination model was studied and if/how this would link to the laminar structure observed in the monkey data. Further, it is unclear how comparable tortuosity/max deviation values are across species, considering the differences in data quality and relative resolution, given that the presented results show these values are very modality-dependent.

R1.8: We have clarified the motivation for including the mouse part of the study in both the Introduction and the Results sections.

The Introduction section (page 5): “Furthermore, using a mouse model of focal demyelination induced by cuprizone (CPZ) treatment, we investigate the inflammation-related influence on axonal organisation. This is achieved through the same structure tensor-derived micro-anisotropy and tractography streamline metrics.”

The Results section (page 15): “Finally, we investigated the organisation of fasciculi in both healthy mouse brains and a murine model of focal demyelination induced by five weeks of cuprizone (CPZ) treatment. This allowed for the exploration of the disease-related influence on axonal organisation, particularly under inflammation-like conditions with high glial cell density at the demyelination site (He et al. 2021). The experimental setup for DESY and ESRF is similar to that described for the monkey, with the exception that we did not perform dMRI and synchrotron imaging on the same brains, and only collected MRI data for healthy mouse brains. This approach allowed us to apply the same structure tensor and tractography streamline analysis used previously, but in a healthy versus disease comparison, demonstrating the methodology’s ability to provide insights into pathological conditions.”

Across species, the comparison of tortuosity and maximum deviation must be approached with caution. On one hand, we observe a comparable influence of the extra-axonal environment in both the monkey and mice, as discussed in the section “Sources to the non-straight trajectories of axon fasciculi.” On the other hand, the anatomical scale and relative image resolution are significant factors, as correctly pointed out. In the mouse, for instance, the measures are influenced by white matter pathway macroscopic effects, making cross-species comparison challenging to perform in a normalised way.

The limitations section of the Discussion (page 28) has been updated to reflect this: ”A limiting consequence of having samples imaged at differing anatomical scales is that certain measures become inherently hard to compare in a normalised way. The tractography-based metrics—tortuosity and maximum deviation—serve as good examples of this resolution and FOV dependence. In the ESRF samples, the anatomical scale was at the level of individual axons, and the streamline metrics primarily reflect micro-scale effects from the extra-axonal environment, such as the influence of cells and blood vessels. In comparison, the larger anatomical scale in the DESY samples represents the level of fasciculi and above, with metrics influenced by macroscopic effects, such as the bending of the CC pathway. Both scales are interesting and can provide valuable insights in their own right, but caution is required when comparing the numbers, especially for cross-species studies where there is a significant difference in brain volume ratios.”

The paper introduces a new method of "scale-space" parameters for structure tensors. Since, to my understanding, this is the first description of the method, some simple validation of the method would be welcomed. Further, the same scale parameters are not used across monkeys and mice, with a larger kernel used in mice (Table 2) which is surprising given their smaller brain size. Some explanation would be helpful.

R1.9: We have expanded the description of the scale-space structure tensor approach in the Methods section. Specifically, we have elaborated on the empirical process used to select the scale-space parameters shown in Table 2 and explained why multiple scales were applied only to the monkey samples scanned at ESRF (see Table 2, sample IDs 2 and 3) but not to the other datasets. Additionally, we have added a supplementary figure to assist in illustrating the concept.

**Reviewer #2 (Public Review):**
Summary:In this work, the authors combine diffusion MRI and high-resolution x-ray synchrotron phase-contrast imaging in monkey and mouse brains to investigate the 3D organization of brain white matter across different scales and species. The work is at the forefront of the anatomical investigation of the human connectome and aligns with several current efforts to bridge the resolution gap between what we can see in vivo at the millimeter scale and the complexity of the human brain at the sub-micron scale. The authors compare the 3D white matter organization across modalities within 2 small regions in one monkey brain (body of the corpus callosum, centrum semiovale) and within one region (splenium of the corpus callosum) in healthy mice and in one murine model of focal demyelination. The study compares measures of tissue anisotropy and fiber orientations across modalities, performs a qualitative comparison of fasciculi trajectories across brain regions and tissue conditions using streamlined tractography based on the structure tensor, and attempts to quantify the shape of fasciculi trajectories by measuring the tortuosity index and the maximum deviation for each reconstructed streamline. Results show measures of anisotropy and fiber orientations largely agree across modalities, especially for larger FOV data. The high-resolution data allows us to explore the fiber trajectories in relation to tissue complexity and pathology. The authors claim the study reveals new common organization principles of white matter fibers across species and scales, for which axonal fasciculi arrange into sheet-like laminar structures.Strengths:The aim of the study is of central importance within present efforts to bridge the gap between macroscopic structures observable in vivo in humans using conventional diffusion MRI and the microscopic organization of white matter tissue. Results obtained from this type of study are important to interpret data obtained in vivo, inform the development of novel methodologies, and expand our knowledge of the structural and thus functional organization of brain circuits.Multi-scale data acquired across modalities within the same sample constitute extremely valuable data that is often hard to acquire and represent a precious resource for validation of both diffusion MRI tractography and microstructure methods.The inclusion of multi-species data adds value to the study, allowing the exploration of common organization principles across species.The addition of data from a murine cuprizone model of focal demyelination adds interesting opportunities to study the underlying biological changes that follow demyelination and how these impact tissue anisotropy and fiber trajectories. These data can inform the interpretation and development of diffusion MRI microstructure models.Weaknesses:The main claim of a newly discovered laminar organization principle that is consistent across scales and species is not supported strongly enough by the data. The main evidence in support of the claim comes from the larger FOV data obtained from the body of the corpus callosum in the monkey brain. A laminar organization principle is partially shown in the centrum semiovale in the monkey brain and it is not shown in mice data. Additionally, the methods lack details to help the correct interpretation of these findings (e.g., how were these fasciculi defined?; how well do they represent different axonal populations?; what is the effect of blood vessels on the structure tensor reconstruction?; how was laminar separation quantified?) and the discussion does not provide a biological background for this organization. The corpus callosum sample suggests axons within a bundle of fibers are organized in a sheet-like fashion, while data from the centrum semiovale suggest fibers belonging to different fiber bundles are organized in a sheet-like arrangement. While I acknowledge the challenges in acquiring such high-resolution data, additional samples from different regions in the same animals and from different animals would help strengthen this claim.

R2.1

- how were these fasciculi defined?

In the introduction (page 3), we have clarified our definition of an axon fasciculus: “A fasciculus is a bundle of axons that travel together over short or long distances. Its size and shape can vary depending on its internal organisation and its relationship to neighbouring fasciculi.”

Additionally, we emphasise in the Results section (page 12) that the centroid streamlines are not guaranteed to be actual fasciculi, but rather representations of them. The paragraph now states: “To ease visualisation and quantification, we used QuickBundle clustering(Garyfallidis et al. 2012) to group neighbouring streamlines with similar trajectories into a centroid streamline. This centroid streamline serves as an approximation of the actual trajectory of a fasciculus.”

- what is the effect of blood vessels on the structure tensor reconstruction?

Fair point, that was not clear from our description. The clarification contains two parts. First, the estimation of the structure tensor occurs in all voxels, and in that sense, the blood vessels respond very similarly to axons. Second, when it comes to sample statistics derived from the structure tensor analysis (FA histograms and the FODs), they will have an influence, albeit a small one, given the low volume percentage of the blood vessels within the FOVs. In the monkey samples, segmenting the blood vessels was achievable with little effort, allowing us to exclude their contribution from FA statistics and FODs. To make this clear, we have added a paragraph to the Methods section (page 34) titled “Structure tensor-based quantifications,” reflecting this clarification. Additionally, we have restructured the entire structure tensor methods description (starting on page 32) as part of the reviewer comments in R1.6 and R1.9.

- how was laminar separation quantified?

We have added a clarification in Results section (page 7): “The laminar thickness was determined by manual measurements on laminae visually identified in the 3D volume”.

- discussion does not provide a biological background for this organization.

A good point. Including the biological background is relevant as it supports the laminar organisation of white matter pathways observed in our findings and those of others.

We have added a section on this background in the Discussion (page 24): “We believe our observed topological rule of white matter laminar organisation can be explained by a biological principle known from studies of nervous tissue development. The first axons to reach their destination, guided by their growth cones, are known as “pioneering” axons. “Follower” axons use the shaft of the pioneering axon for guidance to efficiently reach the target region (Breau and Trembleau 2023). Axons can form a fasciculus by fasciculating or defasciculating along their trajectory through a zippering or unzipping mechanism, controlled by chemical, mechanical, and geometrical parameters. Zippering “glues” the axons together, while unzipping allows them to defasciculate at a low angle (Šmít et al. 2017). Although speculative, the zippering mechanism may be responsible for forming the laminar topology observed across length scales. The defasciculation effect can explain our results in the corpus callosum (CC) of monkeys, with laminar structures at low angles (~35 degrees) also observed by (Innocenti et al. 2019; Caminiti et al. 2009), as well as in other major pathways (Sarubbo et al. 2019). In contrast, a fasciculation mechanism may be observed in the mouse CC (0 degrees). If the geometrical angle between two axons is high, i.e., toward 90 degrees, the zippering mechanism will not occur, and the two axons (fasciculi) will cross (Šmít et al. 2017). This supports our and other findings that crossing fasciculi or pathways occur at high angles toward 90 degrees in the fully matured brain (Wedeen et al. 2012). Once myelination begins, the zippering mechanism is lost (Šmít et al. 2017), suggesting that laminar topology is established at the earliest stages of brain maturation.”

- additional samples from different regions in the same animals and from different animals would help strengthen this claim

Reviewer #1 also pointed to the inclusion of additional samples, and this is now discussed as part of the study limitations on page 27 (see also R1.7).

The main goal of the study is to bridge the organization of white matter across anatomical length scales and species. However, given the substantial difference in FOVs between the two imaging modalities used, and the absence of intermediate-resolution data, it remains difficult to effectively understand how these results can be used to inform conventional diffusion MRI. In this sense, the introduction does not do a good enough job of building a strong motivation for the scientific questions the authors are trying to answer with these experiments and for the specific methodology used.

R2.2: Indeed, this is an essential point now emphasised in the introduction, page 3, which now states: ”Despite the limited resolution of dMRI, the water diffusion process can reveal microstructural geometrical features, such as axons and cell bodies, though these features are compounded at the voxel level. Consequently, estimating microstructural characteristics depends on biophysical modelling assumptions, which can often be simplistic due to limited knowledge of the 3D morphology of cells and axons and their intermediate-level topological organisation within a voxel. Thus, complementary highresolution imaging techniques that directly capture axon morphology and fasciculi organisation in 3D across different length scales within an MRI voxel are essential for understanding anatomy and improving the accuracy of dMRI-based models(Alexander et al. 2019).”

Additionally, in the introduction, page 4, we have made the following changes to strengthen the link across modalities, such that it now states: “In the x-ray synchrotron data, we applied a scale-space structure tensor analysis, which allowed for the quantification of structure tensor-derived tissue anisotropy and FOD in the same anatomical regime indirectly detected by dMRI.”

The cuprizone data represent a unique opportunity to explore the effect of demyelination on white matter tissue. However, this specific part of the study is not well motivated in the introduction and seems to represent a missed opportunity for further exploration of the qualitative and quantitative relationship between diffusion MRI and sub-micron tissue information (although unfortunately not within the same brain sample). This is especially true considering the diffusion MRI protocol for mice would allow extrapolation of advanced measures from different tissue compartments.

R2.3: A similar point was raised by Reviewer 1 (R1.8), and we have clarified the motivation for including the healthy mice and the demyelination samples.

**Recommendations for the authors:**

**Reviewer #1 (Recommendations For The Authors):**
Many thanks to the authors for providing open data. This was very helpful when reviewing the manuscript and is a valuable resource for the community.

R1.10: We are happy to share our data with the community. Understanding anatomy in 3D is hard to achieve through still images and animations, so the ability to explore it on your own is quite important. The link to the data repository has been added in the Methods section in the following paragraph: “Due to the size of the data selected, processed image volumes, masks and results are available at https://zenodo.org/records/10458911. Other datasets can be shared on request.“

One confusing element of the paper is that orientations (or axes) do not seem to be consistent across samples/modalities. For example, the green tensors in Figures 3 C and D are tilted up/down in opposite directions and the streamlines in Figure 5A seem opposite (SL) from what we would expect from Figure 2A (SR). Having consistent orientations across modalities and images would help the reader. When colouring tensors (e.g. in Figure 3), the authors could consider a 3D colour scheme (similar to that used by diffusion MRI) rather than colouring by only inclination, as this would provide useful information on whether different laminae have similar orientations, as implied by the tractography in Figure 4.

R1.11: Thank you for spotting the suboptimal consistency between Figures 2, 3, and 5. Figure 2 has been corrected and updated. The left-right direction in the coronal views was not correctly displayed. Additionally, the glyph directions have been updated in Figures 2 and 3.

By default, we use the “standard” RGB colour scheme used in dMRI. However, for the monkey CC— essentially Figure 3—this did not effectively illustrate our findings. We decided to use a different directional colour encoding scheme, which captures the angular deviation from the L-R axis. This was to assist in the visualisation of the inclination angle between the laminars. We have used the same colour scheme for the tensors in Figure 3 to avoid confusion.

On a general note, the standard colour scheme has uniform “colour contrast” in all directions, but when there is only a single dominant direction in the sample, it can make sense to concentrate the colour contrast in that axis.

Results: "even higher FA anisotropy in the micro-tensor domain of 0.997, i.e., the micro (μ)FA (20, 21)." I understand these references lead to a definition of μFA that is based on multiple diffusion tensor encodings which is quite different from that suggested by Kaden. It may be preferable to reference Kaden directly (since I understand this is the method used) to avoid confusion.

R1.12: Correctly spotted, and we now reference the method from Kaden et al. and use the other references elsewhere when relevant.

"and scanned the mouse brain in a whole." - typo?

R1.13: Thank you for spotting the typo. The mouse brain was kept in the skull during MRI scanning, which has been clarified in the Methods section.

The crossing fibre region appears to be sometimes referred to as the centrum semiovale, and other times as the CST. CS seems the better description and keeping this naming consistent would avoid confusion to the reader.

R1.14: Well spotted, thank you. We have replaced the usage of Corticospinal Tract (CST) with centrum semiovale (CS) where relevant.

Direct comments on the text:Abstract: "Individual axon fasciculi exhibited tortuous paths .... in a manner independent of fibre complexity and demyelination"Do statistical comparisons of the various distributions support this? The data shows somewhat increased tortuosity in the CST compared to the CC, and somewhat lower tortuosity in CPZ tissue.

R1.15: The intention of the text was not to point to the comparison of tortuosity, but rather to highlight the maximum deviation. We observe a high probability density of maximum deviations at approximately 5-10 microns in all samples, which corresponds to the size of structures in the extraaxonal environment, such as blood vessels and cells.

Additionally, we understand that the original statement might imply an expectation of a statistical analysis demonstrating independence, which is not the case. To clarify, we have reformulated the sentence in the Abstract (page 2) to address these points: “Fasciculi exhibited non-straight paths around obstacles like blood vessels, comparable across the samples of varying fibre complexity and demyelination.”

Abstract: "A quantitative analysis of tissue anisotropies and fibre orientation distributions gave consistent results for different anatomical length scales and modalities, while being dependent on the field-of-view."To my understanding, the FODs here from different modalities are calculated over different FOVs (in monkeys at least), and FODs are only presented for a single FOV for each modality, meaning it is difficult to separate the effects of modality from FOV. The microscopic anisotropy is also noticeably different across modalities (DESY < ESRF < dMRI).

R1.16: That is a fair point. Our statement was trying to capture too much condensed content to be correctly interpretable. We have reformulated the sentence to state: “Quantifications of fibre orientation distributions were consistent across anatomical length scales and modalities, whereas tissue anisotropy had a more complex relationship, both dependent on the field-of-view”.

While it is true that we only present the ST-derived quantifications – FOD and FA statistics – for a single FOV per modality and sample, the results shown for the ESRF monkey samples (Figures 3 and 4) are a merge of four individually processed volumes. The quantifications of each individual subFOV have now been added as a supplementary figure (Figure S3) to highlight the consistency of the methodology and the effect of shifting the FOV position. In the case of the mouse, we have two volumes from different mice, which also display similar FOD and FA statistics.

Abstract: "Our study emphasises the need to balance field-of-view and voxel size when characterising white matter features across anatomical length scales."This point does not seem very well explored in the paper, rather it is an observation of the limitations of the different imaging modalities. For example, there aren't analyses to compare metrics from highresolution data at different FOVs (i.e. by taking neighbourhoods of different sizes), nor are metrics compared from data at different resolutions and the same FOV.

R1.17: The question is related to R1.16, R1.4, and R1.8, and we have addressed this point in our responses to those comments.

Figure 7 - Taking into account the eigenvalues can be helpful when interpreting the secondary and tertiary eigenvectors of tensors (V2 and V3). It would be interesting to know whether the eigenvalues L2 ~ = L3 are approximately equal (suggesting isotropic diffusion about V1, where the definition of V2 versus V3 isn't very meaningful), or if L2 is noticeably larger than L3 (suggesting anisotropic diffusion about V1, potentially similar to the anisotropic dispersion discussed above).

R1.18: It would be interesting to explore the eigenvalues of the structure tensor in more detail, as has been done for the diffusion tensor. However, we believe this belongs to future work, as such additional detailed methodological analysis would complicate the already complex story. As mentioned in response to R1.10, most processed data has been made publicly available, and the rest can be requested (due to the storage size of the data sets) to perform such additional analysis.

Discussion: "Importantly, our findings revealed common principles of fibre organisation in both monkeys and mice; small axonal fasciculi and major bundles formed sheet-like laminar structures," See above regarding the lack of evidence for laminar structures in mouse data.

R1.19: We have reformulated the text for clarification as part of R1.3. Additionally, we added FOD quantifications to support why we do not observe an apparent laminar organisation in the mouse CC— please see our response to R1.6.

Discussion: "Interestingly, the dispersion magnitude is indicative of fasciculi that skirt around obstacles in the white matter such as cells and blood vessels, and the results are largely independent of both white matter complexity (straight vs crossing fibre region) and pathology." Again, do statistical tests of the various distributions support this?

R1.20: As part of R1.1, we have added statistical tests of significance for the quantifications of how max deviation changes when bending around objects. Indeed, the distributions are not statistically the same, and we do not wish to convey that sentiment, but they are comparable in the object sizes that they detect. As done in the abstract, we have reformulated the sentence to avoid misunderstanding and have replaced “largely independent” with “observed across.”

Discussion: "Tax et al. have demonstrated the calculation of a sheet probability index from diffusion MRI data, which suggested the presence of sheet-like features in the CC"My understanding was that this was observed in crossing fibre regions, such as where fibres projecting with the CC cross the CST, but not the main body of the CC itself. Tax defines sheet structure as "composed of two tracts that cross each other on the same surface in certain regions along their trajectories." Is this a different phenomenon to the laminar structures observed here (where we observe fibres within a single tract being locally organised into laminar structures)?

R1.21: Thank you for pointing our attention to this. We have corrected the section in the Discussion (page 23), so it now states: “Additionally, Tax et al. have demonstrated the calculation of a gridcrossing sheet probability index from diffusion MRI data, which suggested the presence of sheet-like features in a crossing fibre region (Tax et al. 2016), which is in line with our findings in the synchrotron data. Note that the method by Tax et al. only detects sheet-like structures crossing on a grid and does not reveal laminar structures with lower inclination angles, as we observed in the monkey CC.”

Discussion: "We found that FODs were consistent across image resolutions and modalities, but only given that the FOV is the same." See above.

R1.22: As part of our response to R1.6, we quantified the FODs using the ODI and DA indices, which should help support our statement. Nevertheless, we have toned down the statement and reformulated the text as follows: “We found that FODs were comparable across image resolutions and modalities. The observed discrepancies can be attributed to the fact that the FOVs are not exactly matched.”

Discussion: "microscopic FA were highly correlated across modalities."The data shows FA is considerably lower in DESY to ESRF; within modality FA is quite consistent irrespective of tissue region; and differences between the CC and CG shown in ESRF data in mice are not repeated in DESY. It is unclear from the current data if this would lead to a high correlation across modalities. Some evidence would be helpful.

R1.23: This is a fair point; we have not performed a correlation analysis. However, the pattern we observe for the synchrotron samples is as follows: When the anatomical length scale increases (becomes more macroscopic), the FA distribution shifts to lower values. This reflects the scale of information captured with the ST analysis (see also R1.9). Therefore, the most interesting comparison of FA statistics occurs when the resolution and anatomical length scale are approximately the same. The sentence in question has been reformulated to the following: ”Estimates of structure tensor derived microscopic FA show a clear pattern across modalities.”

Discussion: "If so, the (inclination angle) information might serve to form rules for low-resolution diffusion MRI based tractography about how best to project through bottleneck regions, which is currently a source of false-positives trajectories (6)."This is an interesting idea but it is unclear to me how this inclination information would help track through bottlenecks where, by definition, fibres are passing through with the same orientation. Some further explanation would be helpful.

R1.24: We have elaborated on the section in the Discussion (page 23), explaining how this can be used to improve tractography tracing through complex regions: “The reason is that standard tractography methods do not "remember" or follow anatomical organisation rules as they trace through complex regions. Our findings on pathway lamination and inclination angles—low for parallel-like trajectories and high for crossing-like trajectories—can help incorporate trajectory memory into these methods, reducing the risk of false trajectories”.

**Reviewer #2 (Recommendations For The Authors):**
Below I report comments that if addressed I believe would improve the clarity and readability of the manuscript.- Figures 1 and 2 would be more meaningful if combined into one figure. This would allow for a direct visual comparison of the two modalities. If space is needed, I believe the second row of Figure 1 (coronal views of CC) does not add much information. It is often hard to navigate the different orientations of the tissue in the images; thus any effort in trying to help the reader visually clarify would improve readability.

R2.4: We considered the reviewer’s suggestion to merge Figures 2 and 3. However, this made both the figures and the main text additionally complex, so we chose to retain the original figure layout. Secondly, Figure 3 utilises a non-standard directional colormap. Keeping the colormap consistent within each figure is a feature we wish to preserve. In response to R1.11, the figures have been updated to have more consistent orientations for the monkey samples.

In Figure 2, the second row, showing a coronal view of the CC, is essential for comparison with human data in Figure S1. It highlights where we observed the columnar laminar organisation and their inclination angle, as also detected by DTI.

- Figure 4 shows synchrotron data revealing an anterior-posterior component within the centrum semiovale that is not necessarily seen in the dMRI data. Could the authors comment on this?

R2.5: Thank you for pointing this out. We have now addressed this in the Results section (page 10), where we describe the observation in detail: “Interestingly, visual inspection of the colour-coded structure tensor directions in Fig. 4E shows the existence of voxels whose primary direction is along the A-P axis. However, this represents a small enough portion of the volume that it does not appear as a distinct peak on the FOD.“

- The authors claim they observed several purple axons crossing orthogonally in Figure 5c. However, that is not necessarily clear in the figure.

R2.6: We appreciate the feedback. We have now coloured the streamlines of the crossing fasciculi in Figure 5C in red.

- Figure 5 would benefit from adding the color encoding scheme for Figure 5d, as sometimes this is not necessarily consistent.

R2.7: We appreciate the feedback. We have added an indication of the standard directional colour coding to Figure 5D.

- Figure 5d shows interesting data from the complex region. However, it is hard to visualize and it looks like there are not many streamlines traveling entirely I-S? Maybe a different orientation of the sample would help visualization.

R2.8: A similar point was raised by Reviewer 1 (see R1.2). We have added an animation of the scene to assist in the interpretation of the 3D organisation within this complex sample.

- The concept of axon fasciculi is not necessarily immediately clear. Adding an explanation for what the authors refer to when using this term would improve clarity.

R2.9: In the introduction, we now state our conceptual definition of an axon fasciculus as a number of axons that follow each other (see also R2.1).

- The methods do not provide details on how structure tensor FA is measured.

R2.10: Thank you for pointing this out. We have restructured and expanded the structure tensor description in the Methods section (see also R1.9 and R2.1), which now includes the definition of FA.

- Why didn't the authors select the same cc region for both mice and monkeys? It seems this would have increased the strength of the comparison.

R2.11: We agree. The reason lies in the chronology of experiments and the fact that we cannot control where demyelination takes place. We have added a clarifying description in the Methods section (page 31): “Note that several separate beamline experiments were conducted to collect the volumes listed in Table 1. In the first two experiments, samples from the monkey brain were scanned at ESRF and DESY, respectively. The samples from the mouse brain were imaged in two subsequent experiments. Consequently, the location of the identified demyelinating lesion in the cuprizone mice, which cannot be precisely controlled, did not match the location of the CC biopsies in the monkey.”

- While it is mentioned in the results, the methods do not explain how vessel segmentations or cell segmentation in mice was performed and for which datasets it was performed.

R2.12: For the small ROI shown in Figure 6, the labelling was a manual process using the software ITK-SNAP, which has now been clarified in the corresponding figure caption. The generation of ROI masks and blood vessel segmentations involved a combination of intensity thresholding, morphological operations, and manual labelling in ITK-SNAP. This has been clarified in the restructured and expanded description of structure tensor analysis in the Methods section (starting on page 32).

- From the methods it is hard to understand (1) how many mice were used; (2) why dMRI was done on a different sample; (3) whether the same selenium region was selected for both healthy and CPZ animals; (4) how the registration across samples was performed.

R2.13: We appreciate the feedback and have inserted clarifying statements in the relevant parts of the Methods section. (1) The total number of mice included was three: one normal, one cuprizone, and one normal for MRI scanning. (2) The quality of the collected dMRI on the mouse was too poor to use, and it could not be redone as the brain had already been sliced and prepared for synchrotron experiments. (3) The same splenium section was selected for both healthy and cuprizone mice. (4) A paragraph on image registration has been added.

- Diffusion MRI method sections would benefit from additional details on the protocols used.

R2.14: Thank you for pointing this out. We have added more details about the diffusion MRI protocols, including the b-value, gradient strength, and other relevant parameters.